The earliest dipodomyine heteromyid in North America and the phylogenetic relationships of geomorph rodents

Samuels Joshua X. samuelsjx@etsu.edu 1
Calede Jonathan J.-M. 2
Hunt, Jr. Robert M. 3
1 Department of Geosciences, Don Sundquist Center of Excellence in Paleontology, East Tennessee State University , Johnson City, TN , United States of America
2 Department of Evolution, Ecology, and Organismal Biology, Ohio State University—Marion , Marion , OH , United States of America
3 Department of Earth and Atmospheric Sciences, University of Nebraska—Lincoln , Lincoln , NE , United States of America
Abdala Virginia
Electronic publication date: 2023 Mar 8
Publication date: 2023
Volume: 11
Electronic Location ID: e14693
Received 2022 Aug 16; Accepted 2022 Dec 14
Copyright: ©2023 Samuels et al.
Copyright year: 2023
Copyright holder: Samuels et al.
License: This is an open access article distributed under the terms of the Creative Commons Attribution License, which permits unrestricted use, distribution, reproduction and adaptation in any medium and for any purpose provided that it is properly attributed. For attribution, the original author(s), title, publication source (PeerJ) and either DOI or URL of the article must be cited.
License URL: https://creativecommons.org/licenses/by/4.0/

Keywords: Kangaroo rat, Heteromyidae, Mosaic evolution, Geomorpha

Funding: United States National Park Service and startup funds from East Tennessee State University Paleontological Society Norman Newel Award, a College of Arts and Sciences Regional Campus Research and Creative Activity Grant from the Ohio State University, a research grant from the Ohio State University at Marion, startup funds from the Ohio State University, and visiting researcher support from the Don Sundquist Center of Excellence in Paleontology, East Tennessee State University Funding for this research was provided to Joshua X. Samuels by the United States National Park Service and startup funds from East Tennessee State University. Funding to Jonathan Calède was provided by a Paleontological Society Norman Newel Award, a College of Arts and Sciences Regional Campus Research and Creative Activity Grant from the Ohio State University, a research grant from the Ohio State University at Marion, startup funds from the Ohio State University, and visiting researcher support from the Don Sundquist Center of Excellence in Paleontology, East Tennessee State University. The funders had no role in study design, data collection and analysis, decision to publish, or preparation of the manuscript.

==============================
Dipodomyine heteromyids (kangaroo rats and mice) are a diverse group of arid-adapted ricochetal rodents of North America. Here, a new genus and species of a large dipodomyine is reported from early Miocene-aged deposits of the John Day Formation in Oregon that represents the earliest record of the subfamily. The taxon is known from a single specimen consisting of a nearly complete skull, dentary, partial pes, and caudal vertebra. The specimen is characterized by a mosaic of ancestral and highly derived cranial features of heteromyids. Specifically, the dental morphology and some cranial characteristics are similar to early heteromyids, but other aspects of morphology, including the exceptionally inflated auditory bullae, are more similar to known dipodomyines. This specimen was included in a phylogenetic analysis comprising 96 characters and the broadest sampling of living and extinct geomorph rodents of any morphological phylogenetic analysis to date. Results support the monophyly of crown-group Heteromyidae exclusive of Geomyidae and place the new taxon within Dipodomyinae. The new heteromyid is the largest known member of the family. Analyses suggest that large body size evolved several times within Heteromyidae. Overall, the morphology of the new heteromyid supports a mosaic evolution of the open-habitat adaptations that characterize kangaroo rats and mice, with the inflation of the auditory bulla appearing early in the group, and bipedality/ricochetal locomotion appearing later. We hypothesize that cooling and drying conditions in the late Oligocene and early Miocene favored adaptations for life in more open habitats, resulting in increased locomotor specialization in this lineage over time from a terrestrial ancestor.

Introduction

The clade Geomorpha is a diverse group of rodents that includes two extant New World families, the Geomyidae (pocket gophers) and Heteromyidae (pocket mice, kangaroo rats, and their relatives) (Flynn, Lindsay & Martin, 2008). These two families are abundant and important components of modern mammal communities across much of North America, and exhibit both high species diversity and great ecomorphological disparity (see Noftz & Calede, 2022). Geomyidae are the most species-rich group of burrowing rodents in North America today (over 40 living species; Burgin et al., 2018), and were so through much of the Cenozoic (Lacey, Patton & Cameron, 2000; Samuels & Hopkins, 2017; Calede, Samuels & Chen, 2019). The Heteromyidae are represented by about 60 living species that include the bipedal ricochetal Dipodomyinae as well as the quadrupedal saltatory Perognathinae common in deserts and other arid habitats of western North America, and the relatively generalized Heteromyinae, which live in more humid habitats of Mexico, Central America and Southern South America (Genoways & Brown, 1993; Schmidly, Wilkins & Derr, 1993; Williams, Genoways & Braun, 1993; Hafner et al., 2007; Fernández et al., 2014). The best studied geomorph rodents may be the kangaroo rats and mice (Dipodomyinae), which have been the subject of extensive research focused on their locomotor adaptations and hearing (e.g., Hatt, 1932; Howell, 1932; Wood, 1935; Bartholomew & Caswell, 1951; Webster, 1962; Webster & Webster, 1971; Webster & Webster, 1975; Webster & Webster, 1984; Gambaryan, 1974; Brylski, 1993; Lay, Genoways & Brown, 1993; Samuels & Van Valkenburgh, 2008; Alhajeri & Steppan, 2018).

In addition to their modern diversity, the Geomorpha also have a rich fossil record (Wood, 1935; Wahlert, 1991; Korth, 1994; Flynn, Lindsay & Martin, 2008; Calede & Rasmussen, 2020), with well-preserved specimens documenting the morphology of extinct species that have enabled paleoecological inferences (Calede, Samuels & Chen, 2019; Scarpitti & Calede, 2022). However, the evolution of Geomorpha remains relatively poorly understood (Flynn, Lindsay & Martin, 2008), and morphological phylogenies of fossil geomorphs to date (e.g., Wahlert, 1991; Korth, 2008) have been inconsistent with relationships based on well-supported molecular evidence (e.g., Hafner et al., 2007; Fabre et al., 2012; Upham, Esselstyn & Jetz, 2019). Starting with Wood (1935), authors studying fossil taxa have consistently allied Perognathinae with Dipodomyinae based on the central fusion of the lophs of the lower p4 (x-pattern), a character whose evolution is poorly understood. As a consequence, a number of fossil heteromyid taxa known from relatively complete material (like Schizodontomys) that show some cranial similarities with dipodomyines (including inflated auditory bullae; Wahlert, 1985; Wahlert, 1993) have been hypothesized to be closely related to heteromyines or even represent basal heteromyids (Korth, Bailey & Hunt Jr, 1990; Korth, 1997; Flynn, Lindsay & Martin, 2008), based on the fact that the lophs of the p4 fuse lingually and labially, isolating a central enamel basin in those taxa.

Although details of the relationships among heteromyids have varied in recent molecular studies, they consistently place the heteromyid subfamilies Heteromyinae and Perognathinae as sister clades and Dipodomyinae outside of that group (Hafner et al., 2007; Fabre et al., 2012; Upham, Esselstyn & Jetz, 2019). Those studies clearly show that the central fusion of lophs in the p4 of perognathines and dipodomyines either arose twice independently or arose once with a subsequent reversal in heteromyines. Regardless of the exact evolutionary scenario, tooth morphology should not be assumed to be free from convergence in heteromyids and should not be considered more reliable than cranial morphology in determining relationships of fossil taxa. Parallelism in dental morphology is certainly common in geomyoids (Wood, 1935; Munthe, 1981; Brylski, 1993; Hafner, 1993) and no study to date has provided a detailed morphological phylogeny of the group with a broad sampling of both extant and fossil taxa.

Here, a new, large heteromyid is described from the early Miocene of Oregon. A skull, dentary, partial pes, and caudal vertebra were recovered in situ from the Johnson Canyon Member of the John Day Formation, and show excellent preservation of morphological attributes. In addition to the description of the new taxon, a new morphological phylogeny of living and fossil geomorph rodents is provided, including many basal heteromyid and geomyoid taxa that have never been included in a cladistic analysis. The detailed study of both extant heteromyid species and well-represented fossil taxa employed here informs heteromyid phylogeny and improves understanding of the evolution of geomorph rodents, an important group of North American small mammals. Furthermore, the phylogenetic framework provides the basis for future improvements of the phylogenetic systematics of Geomorpha and the phylogenetic comparative analysis of ecological data within this clade.

Material and Methods

Studied material and comparative methods

A large comparative sample of geomorph rodents was examined quantitatively and qualitatively (Table 1), including all extant heteromyid genera and a wide range of fossil geomorph taxa. The examination of specimens was supplemented with data from published descriptions and photographs included in a wide range of studies. A full listing of the taxa and specimens studied for qualitative comparisons is provided in Table S1. The nomenclature used for the description of cranial structures follows Wahlert (1985) and Wahlert (1991). Detailed morphological descriptions of cranial structure, foramina, and bone sutures are provided for Dipodomys in Howell (1932), Heteromys, Perognathus, and Microdipodops in Wahlert (1985), Schizodontomys in Korth, Bailey & Hunt Jr (1990), Cupidinimus in Korth (1998), and Eochaetodipus and Mioperognathus in Korth (2008). In this study cranial foramina and bone sutures are illustrated similarly in a new taxon, as well as Bursagnathus aterosseus and Proheteromys latidens, which were described in detail but not illustrated by Korth & Samuels (2015). The dental nomenclature used in comparative descriptions follows that of Wood & Wilson (1936) with modifications for heteromyids from (Korth, 1997: Fig. 1). Descriptions of teeth in this study use the terms medial and lateral, rather than lingual or labial/buccal (in contrast to Wood & Wilson, 1936). Upper teeth are designated by capital letters, lower teeth by lower-case letters (e.g., M1, m1). Specimens were photographed using either a Nikon D810 DSLR camera with an AF-S Micro Nikkor 60 mm lens or a DinoLite Edge AM4815ZT digital microscope camera.

Table 1 Comparative sample of modern and fossil species used in the study, including OTUs included in the phylogenetic analysis and references for morphological information.

A full listing of modern and fossil specimens is provided in Table S1.

Genus	Species	References	Source	
Outgroup-Eomyidae			
Paradjidaumo	trilophus	Wahlert (1978), Korth (1980) and Korth (2013)	Specimens	
Outgroup-Ischyromyidae			
Paramys	delicatus	Matthew (1910), Wood (1962), Wahlert (1974), Wahlert (1991), Rybczynski (2007) and Bertrand, Amador-Mughal & Silcox (2016)	Literature	
Heliscomyidae			Literature	
Heliscomys	ostranderi	Korth, Wahlert & Emry (1991) and Asher et al. (2019)	Literature	
Heliscomys	vetus	Korth (1995)	Literature	
Megaheliscomys	mcgrewi	Korth (2007)	Literature	
Florentiamyidae			Literature	
Ecclesimus	tenuiceps	Galbreath (1948), Galbreath (1961), Black (1961) and Korth (1989)	Literature	
Florentiamys	kingi	Wahlert (1983)	Specimens	
Hitonkala	andersontau	Korth (1993b)	Specimens	
Kirkomys	nebraskensis	Wahlert (1984) and Korth & Branciforte (2007)	Literature	
Sanctimus	simonisi	Wahlert (1983)	Specimens	
Sanctimus	stouti	Wahlert (1983)	Specimens	
Sanctimus	stuartae	Rensberger (1973b)	Specimens	
Geomyoidea incertae sedis			
Balantiomys	oregonensis	Gazin (1932)	Specimens	
Harrymys	irvini	Wahlert (1991) and Korth (1997)	Specimens	
Mioheteromys	amplissimus	Korth (1997)	Specimens	
Mojavemys	galushai	Korth & Chaney (1999)	Specimens	
Phelosaccomys	neomexicanus	Korth & Chaney (1999)	Specimens	
Proharrymys	schlaikjeri	Black (1961)	Specimens	
Proheteromys	latidens	Wood (1932) and Korth & Samuels (2015)	Specimens	
Tenudomys	dakotensis	Korth (1993a)	Specimens	
Trogomys	rupinimenthae	Reeder (1960)	Specimens	
Heteromyidae-Dipodomyinae			
Cupidinimus	nebraskensis	Wood (1935) and Korth (1979)	Specimens	
Dipodomys	merriami	Wahlert (1985) and Anderson, Weksler & Rogers (2006)	Specimens	
Eodipodomys	celtiservator	Voorhies (1975)	Literature	
Microdipodops	megacephalus	Wahlert (1985) and Anderson, Weksler & Rogers (2006)	Specimens	
Prodipodomys	sp.	AMNH F:AM 87427	Specimens	
Heteromyidae-Heteromyinae			
Heteromys	desmarestianus	Dowler & Genoways (1978), Wahlert (1985) and Anderson, Weksler & Rogers (2006)	Specimens	
Heteromys	pictus	Wahlert (1983) and Anderson, Weksler & Rogers (2006)	Specimens	
Heteromyidae-Perognathinae			
Bursagnathus	aterosseus	Korth & Samuels (2015)	Specimens	
Chaetodipus	artus		Specimens	
Chaetodipus	hispidus		Specimens	
Eochaetodipus	asulcatus	Korth (2008)	Literature	
Mioperognathus	willardi	Korth (2008)	Specimens	
Perognathus	furlongi	Gazin (1930)	Specimens	
Perognathus	amplus	Wahlert (1985)	Specimens	
Heteromyidae				
Schizodontomys	amnicolus	Korth, Bailey & Hunt Jr (1990) and Korth (1997)	Specimens	
Schizodontomys	harkseni	MacDonald (1970), Rensberger (1973a) and Munthe (1981)	Specimens	
Schizodontomys	sulcidens	Rensberger (1973a) and Korth (1997)	Specimens	
Geomyidae-Entoptychinae			
Entoptychus	sp.	UCMP 65251; Wahlert (1985) ; Rensberger (1971)	Specimens	
Gregorymys	formosus	Wahlert & Souza (1988); Rensberger (1971)	Specimens	
Pleurolicus	sulcifrons	Rensberger (1973a) and Souza (1989)	Specimens	
Geomyidae –Geomyinae			
Cratogeomys	merriami	Wahlert (1985)	Specimens	
Geomys	arenarius	Asher et al. (2019)	Specimens	
Parapliosaccomys	cf. P. oregonensis	Shotwell (1967) and Kelly & Lugaski (1999)	Specimens	
Pliosaccomys	dubius	Wilson (1936)	Specimens	
Thomomys	talpoides		Specimens	

Figure 1 John Day Formation stratigraphic section at Johnson Canyon East (UCMP Loc. V-6432), near Kimberly OR (redrawn and modified from Hunt Jr & Stepleton, 2004).

The star indicates the level at which UNSM 27016 was recovered in situ.

Figure 2 Holotype specimen of Aurimys xeros from the Johnson Canyon Member of the John Day Formation.

UNSM 27016, skull with right incisor and P4 to M2, and left P4 to M3, left dentary with incisor and p4 to m3, partial metatarsals and proximal phalanges, caudal vertebra. (A–E) Skull: (A) dorsal view, (B) ventral view, (C) enlarged view of upper dentition, (D) right lateral view, (E) left lateral view; (F–H) left dentary: (F) lateral view, (G) dorsal view, (H) enlarged view of lower dentition. Scale bars equal one cm for A–B, D–G, and 2 mm for C and H.

Measurements to the nearest 0.01 mm were taken directly from specimens using Mitutoyo Absolute Digimatic calipers or from digital photographs using ImageJ software (Rasband, 2018). Dimensions of cheek teeth were measured at the occlusal surface following Carrasco (2000). Dental measurements include maximum length (anteroposterior) and width (mediolateral) of the incisors and cheek teeth (fourth premolar and first to third molar), as well as the lengths of the upper and lower diastemata and cheek tooth rows. Cranial measurements largely follow measurements of heteromyid species used in Korth & Samuels (2015). Measurements of pes elements include length and mediolateral width following Samuels & Van Valkenburgh (2008). Definitions for cranial, dental, and postcranial measurements are provided in Table S2 and a complete listing of measurements for all specimens are provided in Table S3. Crown heights of heteromyids were categorized into brachydont, mesodont, hypsodont, or hypselodont based on data derived from Samuels & Hopkins (2017) and the NOW Database of Fossil Mammals (The NOW Community, 2017). First and last appearance dates for fossil taxa were also gathered from those and other sources (Samuels & Hopkins, 2017; The NOW Community, 2017; Williams et al., 2018). These data are used to examine body size and model the evolution of crown height in Heteromyidae.

Institutional abbreviations are as follows: AMNH, American Museum of Natural History, New York City, New York; CM, Carnegie Museum of Natural History, Pittsburgh, Pennsylvania; ETMNH, East Tennessee State University Museum of Natural History, Johnson City, Tennessee; JODA, John Day Fossil Beds National Monument, Oregon; LACM, Natural History Museum of Los Angeles County, Los Angeles, California; SDSM, South Dakota School of Mines and Technology Museum of Geology, Rapid City, South Dakota; UCLA, University of California, Los Angeles, Donald R. Dickey Collection, Los Angeles, California; UCMP, University of California Museum of Paleontology, Berkeley, California; UCMVZ, University of California Museum of Vertebrate Zoology, Berkeley, California; UNSM, University of Nebraska State Museum, Lincoln, Nebraska; USNM, National Museum of Natural History, Smithsonian Institution, Washington DC; UWBM, University of Washington Burke Museum, Seattle, Washington.

Three-dimensional data acquisition

The specimen of the new species described herein was scanned by microfocus X-ray computed tomography (micro-CT) using a Skyscan model 1273 (https://www.bruker.com) at East Tennessee State University. The scans were processed using NRecon and Object Research Systems, Inc. (ORS) (2022) (https://www.theobjects.com/dragonfly). The CT image series and mesh models of the skull and dentary are available at MorphoSource (skull CT image series = https://doi.org/10.17602/M2/M460741, skull mesh model = https://doi.org/10.17602/M2/M460708, dentary CT image series = https://doi.org/10.17602/M2/M468758, dentary mesh model = https://doi.org/10.17602/M2/M468765.

Phylogenetic analysis and ancestral character state reconstruction

The relationships of Geomorpha have been the subject of several phylogenetic studies, including analyses of both molecular and morphological data. Several recent molecular and morphological studies have suggested that Heteromyidae is a paraphyletic group with Geomyidae nested within (DeBry, 2003; Fabre et al., 2012; Asher et al., 2019), but the most comprehensive molecular study of the group (Hafner et al., 2007) found the extant members of the family to be monophyletic. Several recent molecular phylogenies have also shown variable results across analyses, with Upham, Esselstyn & Jetz (2019) alternatively finding the paraphyly of Heteromyidae or monophyly of the family with Geomyidae as their sister group. Within Heteromyidae, relationships among subfamilies are consistent across recent molecular studies. Heteromyinae and Perognathinae are sister clades and Dipodomyinae is outside of that group (Hafner et al., 2007; Fabre et al., 2012; Upham, Esselstyn & Jetz, 2019). These recent molecular phylogenetic studies provide a framework for interpreting the relationships of the group in morphological phylogenetic analyses. The goals of phylogenetic analysis in this study were to: (1) evaluate the relationships of the new fossil taxon to known clades, (2) provide an overall framework for the phylogenetic relationships of living and extinct geomorph rodents, and (3) to shed light on relationships within the family Heteromyidae.

Matrix assembly

A matrix was built including 96 characters and 47 taxa (Tables S4 and S5, Data S1 and S2). The ingroup includes 45 Operational Taxonomic Units (OTUs), which encompass all but one major family within Geomorpha (Table 1 and S5, Data S1 and S2): Heliscomyidae, Florentiamyidae, Heteromyidae, and Geomyidae; only the Jimomyidae are not included (Flynn, Lindsay & Martin, 2008). The choice of such broad sampling enables the exploration of the phylogenetic relationships across Geomorpha. It also helps establish a framework for future studies of stem taxa and species that do not belong to one of the two extant families of geomorph rodents, Heteromyidae and Geomyidae (Flynn, Lindsay & Martin, 2008). Indeed, Flynn, Lindsay & Martin (2008) emphasized the importance of developing a phylogenetic framework of geomorph rodents that includes stem taxa and problematic genera like Harrymys, Tenudomys, Proheteromys, and mojavemyines.

Because the focus of this study was on the systematic paleontology of the new heteromyid taxon from Oregon, this family was extensively sampled. The sample includes representatives from all proposed subfamilies of heteromyids including “Mioheteromyinae”, Heteromyinae, Perognathinae, Mojavemyinae, Harrymyinae, and Dipodomyinae, as well as “basal heteromyids” like Proheteromys and Trogomys, and also taxa that have tentatively been assigned to various heteromyid clades like Schizodontomys (Flynn, Lindsay & Martin, 2008: 436; Korth, 1997; Korth & Chaney, 1999; Wahlert, 1985; Wahlert, 1991). Based on the inflation of the tympanic region of the new material from the John Day Formation, the sample was chosen to include representatives of every single genus of heteromyid of the subfamily Dipodomyinae. Thus, the sample comprises as many as 26 heteromyid taxa (depending on the higher-level taxonomy of fossil taxa).

Characters were scored for eight different geomyid species including crown-group geomyines, stem geomyines, and entoptychines. This decision was made to further test the monophyly of Heteromyidae excluding Geomyidae. The rest of the geomorph sampling was achieved by including three species of heliscomyids and seven different species of florentiamyids representing five genera; all florentiamyid genera except Fanimus in fact. Two different rodents were chosen as outgroups: the ischyromyid Paramys delicatus, one of the oldest rodents in North America, and Paradjidaumo trilophus, a member of the Eomyidae, a possible sister taxon to Geomorpha (Anderson, 2008; Flynn, 2008; Wahlert, 1991). Although most OTUs are identified to the species level, two OTUs are specimens that may represent new species and have not yet be formally described: AMNH F:AM 87427 is a partial skeleton of Prodipodomys, which may be referable to Prodipodomys kansasensis (Hibbard, 1937) or represents a new taxon; UCMP 65251 is a partial skull with associated dentaries of Entoptychus that is distinct from all published species (Wahlert, 1985; J. Calede 2021 pers. obs.). Focus was placed on taxa with known skulls for which many cranial characters could be scored because numerous cranial characters have been demonstrated to be informative of phylogenetic relationships within Geomorpha (e.g., Wahlert, 1985; Wahlert, 1991; Korth, 1993a; Korth, 1993b; Jiménez-Hidalgo, Guerrero-Arenas & Smith, 2018; Calede & Rasmussen, 2020).

Characters (Table S4) were taken from prior analyses of the phylogenetic relationships with Heteromyidae (Hafner, 1978; Anderson, Weksler & Rogers, 2006; Korth, 2008), the only other phylogenetic analysis published for Geomorpha (Wahlert, 1991), a prior analysis of the relationships within Entoptychinae (Calede & Rasmussen, 2020), a broad scale analysis of rodent relationships including select extant and fossil geomorphs (Asher et al., 2019), systematic descriptions and revisions of select taxa included in this analysis (Korth, Wahlert & Emry, 1991; Korth, 1989; Korth, 1993a; Korth, 1993b; Korth, 1997; Flynn, Lindsay & Martin, 2008), past hypotheses of character evolution (Wahlert, 1985), and personal observations of the OTUs studied. Analyses primarily included cranial characters and exclude many dental characters because: (1) some might be highly affected by homoplasy (e.g., the x-pattern of the p4) (see also Dalquest & Carpenter, 1986; Korth, Boyd & Person, 2019), (2) some are associated with ecological traits (e.g., hypsodonty), (3) some are autapomorphic (e.g., presence of a medial sulcus on the upper incisor), and/or (4) they are difficult to score in many taxa because of tooth wear. Only unambiguous parsimony-informative characters were retained. On average, a character could be scored in 34 of the 47 OTUs. The most poorly known OTU (Perognathus furlongi) is only 29% scored; over 38% of ingroup OTUS are 85% or better scored. The Johnson Canyon fossil is 75% scored. The matrix as a whole is 72.7% filled.

Parsimony analysis

The resulting matrix was analyzed using parsimony in PAUP 4.0a build 169 (Swofford, 2002). Multistate characters were treated as polymorphic and 23 characters representing morphological continua were ordered (Table S4). All characters were initially input with equal weights. The analysis was constrained by enforcing several monophylies. This decision was made based on prior analyses, particularly molecular analyses (e.g., Alexander & Riddle, 2005; Fabre et al., 2012). Three genera, all extant, were constrained within the family Heteromyidae: Perognathus, Chaetodipus, and Heteromys. The two subfamilies of pocket gophers (Geomyidae) were constrained based on prior analyses of modern and fossil taxa (Wahlert, 1991; Spradling et al., 2004; Bradley, Thompson & Chambers, 2010; Fabre et al., 2012; Calede & Rasmussen, 2020). Finally, monophyly was enforced for two extinct families of geomorphs: Florentiamyidae and Heliscomyidae (Korth, 1993b; Wahlert, 1983; Wahlert, 1984; Wahlert, 1991; Flynn, Lindsay & Martin, 2008).

A heuristic search was performed using the parsimony criterion and the tree-bisection and reconnection branch swapping to find 1,000 tree replicates via random addition. After this initial analysis, the characters were reweighed using the rescaled consistency index (Farris, 1969; Farris, 1989) following prior analyses of rodent phylogenetics (e.g., Rybczynski, 2007; Van Daele et al., 2007; Calede, 2014; Le Grange et al., 2015). This decision was made to reduce the effect of homoplasies in the parsimony analysis (Farris, 1969). The rescaled dataset was run using a heuristic search with trees added randomly via stepwise addition. One thousand replicates were run with the tree-bisection-reconnection algorithm for branch-swapping, enforcing the constraints described above. All most-parsimonious trees from this second analysis were used to generate a consensus tree, which was visualized in FigTree 1.4.4 (Rambaut, 2018). The nexus file for the analysis is provided in Data S1.

Bayesian analysis

A Bayesian phylogenetic analysis was run using MrBayes 3.2 (Ronquist & Huelsenbeck, 2003). Paramys delicatus was used as the outgroup for this analysis. The set of characters, ordering of characters, and clade constraints were kept identical to those used in the parsimony analysis. The gamma parameter was set to allow characters to evolve at different rates and used two replicate runs with four chains (three heated and one cold) run for one million generations sampling every one hundred generations. Tracer 1.7.1 was used to check for stationarity and used a burn-in of 25% and FigTree 1.4.4 to generate the maximum credibility tree. The associated nexus file is provided in Data S2.

Ancestral character state reconstruction

The consensus tree of the parsimony analysis was used to reconstruct ancestral character states for hypsodonty within Heteromyidae because all most parsimonious trees showed the exact same topology for Heteromyidae. The tree was time-calibrated using the approach of Brusatte et al. (2008) implemented in the package strap 1.4 (Bell & Lloyd, 2015). Maximum likelihood implemented in the ape 5.6-2 package (Paradis, Claude & Strimmer, 2004) was used to reconstruct ancestral character states at the nodes.

Geological setting

Widely distributed through eastern and central Oregon, the John Day Formation includes an incredibly complex series of strata, which consist primarily of volcaniclastic sedimentary rocks and airfall tuffs (Fisher & Rensberger, 1972; Robinson, Brem & McKee, 1984; Bestland & Retallack, 1994; Retallack, Bestland & Fremd, 2000; Albright III et al., 2008; McClaughry et al., 2009). The stratigraphy of the John Day Formation has been studied extensively, most recently by Hunt Jr & Stepleton (2004) and Albright III et al. (2008). The result is a detailed litho- and chronostratigraphy for the formation with radioisotopic and paleomagnetic calibration (Hunt Jr & Stepleton, 2004; Albright III et al., 2008). The John Day Formation (as currently recognized) consists of seven members spanning the late middle Eocene to early Miocene, about 39 to 18 Ma (Hunt Jr & Stepleton, 2004; Albright III et al., 2008).

The specimen described here (UNSM 27016) was recovered with all elements associated, within a small block collected in situ in Johnson Canyon (UCMP V6432) in Grant County, Oregon. Strata at the site are part of the Johnson Canyon Member of the John Day Formation (Hunt Jr & Stepleton, 2004; Albright III et al., 2008), which includes a prominent tuff (“Across the River Tuff”) near its base. The block containing the specimen was recovered from a silt lens within the pebble conglomerate and tuffaceous sandstone at an elevation of 2040 ft (Fig. 1), visible in the stratigraphic section from Hunt Jr & Stepleton (2004), found 80 ft above the “Across the River Tuff”. Both Hunt Jr & Stepleton (2004) and Albright III et al. (2008) interpret the Johnson Canyon Member to have been deposited prior to the Rose Creek Member, which represents the stratigraphically highest member of the John Day Formation. The Rose Creek Member at Picture Gorge 36 has been biostratigraphically dated at ∼18.8–18.2 Ma (Hunt Jr & Stepleton, 2004) and magnetostratigraphically dated to ∼18.7–18.5 Ma or 18.1–17.6 Ma (Albright III et al., 2008). The fauna from the Johnson Canyon Member includes mammals found in late or latest Arikareean age faunas from the Great Plains and lacks any mammals of early Hemingfordian age (Hunt Jr & Stepleton, 2004).

Many tuffs have been radioisotopically dated from the John Day Formation, in the past using 40Ar/39Ar single-crystal laser-fusion dating of sanidine (Albright III et al., 2008). The most relevant date to this study is the “Across the River Tuff” dated to 22.6 ± 0.13 Ma (Swisher III in Fremd, Bestland & Retallack, 1994; Albright III et al., 2008). That older published 40Ar/39Ar date has been recalibrated relative to the new Fish Canyon Tuff sanidine interlaboratory standard of 28.201 Ma (Kuiper et al., 2008), which was done using the ArAR application of Mercer & Hodges (2016) (available at http://group18software.asu.edu). That recalibration yielded a date of 22.746 ± 0.146 Ma for the “Across the River Tuff”. This radiometric date, along with biostratigraphic and magnetostratigraphic dating of overlying strata, allows us to infer the age of the specimen described here, definitively indicating an early Miocene age, either late or latest Arikareean.

Systematic paleontology

Order Rodentia Bowdich, 1821	
Family Heteromyidae Gray, 1868	
Subfamily Dipodomyinae Gervais, 1853	
Aurimys, new genus	

Type and Only Species.—Aurimys xeros, new species.

Diagnosis.—Auditory bullae with ventral and lateral inflation. Mastoid with dorsal, lateral, and posterior portions inflated. Buccinator and masticator foramina fused. Distinct swelling present at posteroventral border of the infraorbital foramen (unlike other fossil and extant dipodomyines with the exception of Dipodomys). Premaxillary-maxillary suture crosses the midline of the palate 1/3rd the distance from the posterior margin of the incisive foramen (unlike other fossil and extant dipodomyines). Posterior border of the maxillary root of the zygomatic arch lies lateral to P4 and anterior to the posterior end of the nasals and premaxillae. Posterior end of the nasals extends farther posteriorly than the premaxillae (unlike other fossil and extant dipodomyines). Interparietal constricted due to auditory bulla expansion. No supraorbital bony flange present (unlike Cupidinimus and extant dipodomyines). Upper incisors lack a central groove. Premolars and molars lack chevrons (unlike other fossil and extant dipodomyines).

Range.—Early Miocene (late or latest Arikareean) of Oregon.

Etymology.—Greek, auri, ear: in reference to the inflated auditory bulla of dipodomyines; Greek, mys, mouse.

Aurimys xeros, new species	
(Figs. 2–7; Tables 2–4)	

Type Specimen.—UNSM 27016, associated skull, left dentary, partial pes (3 incomplete metatarsals and 2 proximal phalanges), 2 caudal vertebrae.

Horizon and Locality.—Johnson Canyon (UCMP V6432), Grant County, Oregon, Johnson Canyon Member, John Day Formation.

Age.—Early Miocene, Late Arikareean (Ar3 or Ar4). UNSM 27016 was collected from above the “Across the River Tuff” (22.746 ± 0.146 Ma, Fremd, Bestland & Retallack, 1994; Albright III et al., 2008, recalibrated as described above) and the Johnson Canyon Member lies stratigraphically below the Rose Creek Member, which has been biostratigraphically dated at ∼18.8–18.2 Ma (Hunt Jr & Stepleton, 2004) and magnetostratigraphically dated to ∼18.7–18.5 Ma or 18.1–17.6 Ma (Albright III et al., 2008). The fauna of the Johnson Canyon Member supports a late to latest Arikareean age (Hunt Jr & Stepleton, 2004).

Diagnosis.—Same as for genus.

Etymology.—Greek, xeros, arid: in reference to the arid habitats favored by dipodomyines.

Description.—The holotype skull (UNSM 27016) is nearly complete (Figs. 2–5), though small fractures occur throughout the specimen (Figs. 2–3). The anterior portion of the rostrum is broken, missing the anterior ends of both nasals, most of the right premaxilla, and the right I1 (Figs. 3A–3B, 3D and 3F). The left I1 is incomplete, the left M3 is missing, and the anterolateral portion of the RP4 is broken, but the other upper dentition is intact if heavily worn (Figs. 2C and 3B). Both zygomatic arches are incomplete, with the left one more complete than the right one (Figs. 2A–2B, 2D–2E, 3A–3B, 3E–3F). The left jugal is missing, as is the right one, but the maxillary root of the right zygomatic arch is also missing the lateral and posterior extensions; the left maxillary root is more complete, but somewhat anteriorly and laterally displaced due to taphonomy. The right auditory bulla is nearly intact (Fig. 3C), missing only a small portion of the tympanic and mastoid along the posterior/ventral margin of the external acoustic meatus; the left bulla is less complete, missing portions of the inflated posterolateral part of the tympanic and mastoid dorsal, posterior, and ventral to the external acoustic meatus. The posterior part of the left palatine is broken along with the pterygoids and medial portion of the alisphenoids (Figs. 2B, 3B). Portions of the basioccipital are missing, but the occipital condyles are intact. The associated left dentary is nearly complete, but the ventral margin ventral to the diastema is broken, with most of the digastric process missing (Figs. 2F, 2G, 3G, 3H, 3I and 6). The coronoid process is missing and the angular process is broken posteriorly, but the articular process is complete. The lower dentition is completely intact, but heavily worn (Figs. 2H and 3G). Preserved on top of the skull is a portion of a pes, including fragments of three metatarsals, which are all missing the proximal end and possess incomplete distal ends, as well as two complete proximal phalanges (Figs. 2A, 3A and 4). An incomplete caudal vertebra is also associated with the specimen, which prior to preparation rested above the left orbit and frontal, overlying the distal end of the left-most proximal phalanx.

Figure 3 Three-dimensional reconstructions of UNSM 27016, holotype specimen of Aurimys xeros, based on micro-CT data.

(A) Dorsal view of skull, (B) ventral view of skull, (C) posterior view of the skull, (D) anterior view of the skull, (E) left lateral view of skull, (F) right lateral view of the skull with origin for anterior zygomatic muscle highlighted by a red dashed line, (G) occlusal view of dentary, (H) medial view of dentary, (I) lateral view of dentary. Scale bar equals 1 cm. Skull mesh model: https://doi.org/10.17602/M2/M460708, Dentary mesh model: https://doi.org/10.17602/M2/M468765.

Figure 4 Morphology of the dorsal and ventral views of the skulls of Aurimys xeros (UNSM 27016), Bursagnathus aterosseus (UCMP 56279), and Proheteromys latidens (UCMP 150688).

Selected anatomical features are highlighted, abbreviations are as follows: aaf, anterior alar fissure; fm, foramen magnum; foa, accessory foramen ovale; fo, foramen ovale; ifo, infraorbital foramen; in, incisive foramen; ju, jugular foramen; mbf, fissure medial to bulla; paf, posterior alar fissure; pgl, postglenoid foramen; pom, posterior maxillary foramen; ppl, posterior palatine foramen; st, stapedial foramen.

Figure 5 Morphology of the lateral views of the skulls of Aurimys xeros (UNSM 27016), Bursagnathus aterosseus (UCMP 56279), and Proheteromys latidens (UCMP 150688).

Selected anatomical features are highlighted, abbreviations are as follows: asq, anterior squamosal foramen; bu, buccinator foramen; eam, external acoustic meatus; eth, ethmoid foramen; foa, accessory foramen ovale; ifo, infraorbital foramen; msc, masticator foramen; paf, posterior alar fissure; rp, rostral perforation; tem, temporal foramen; uaf, unossified area between alisphenoid and frontal bones; uml, unossified area between maxillary and lacrimal bones.

The most obvious feature of the skull is the conspicuous inflation of the auditory bullae and surrounding mastoid (Figs. 2–5). The tympanic bone of the bulla is a single lamina, and the auditory bulla displays anterior, ventral, and lateral inflation (Figs. 2–4 and 7A). Similarly, the mastoid portion of the squamosal bone shows dorsal, posterior, and lateral inflations (Figs. 2–4 and 7A). This results in the cranium being dominated by the ear regions, with each nearly as large as the braincase. The anteromedial bullar processes are present, with expansion of the bullae making them nearly meet along the midline of the ventral aspect of the skull (Figs. 2B, 3B and 4). The convergence of the bullae is not as pronounced on the dorsal aspect of the skull (Figs. 2A, 3A and 4). Posterior inflation is such that the ear regions extend posterior to the occipital bone. The external acoustic meatus is large and round (Figs. 2D–2E, 3F and 5).

Many cranial foramina are preserved on the holotype skull (Figs. 2B, 2D, 2E, 3–5). The incisive foramina are incomplete, but the premaxillary-maxillary suture crosses the palate about 1/3rd of the distance from the posterior margin of the foramen (Figs. 2B and 4). A large rostral perforation is present anterior to the infraorbital foramen, and below the anterior margin of the maxillary root of the zygomatic arch (Figs. 2D–2E, 3E and 5). The infraorbital canal is depressed into the rostrum and there is a clear unossified area between the maxillary and lacrimal bones. The posterior maxillary notch is closed. The anterior alar fissure lies within the alisphenoid and arises far posterior to the M3 (Figs. 2B and 4). The posterior alar fissure is present and joined with the foramen ovale, which is bounded posteriorly by the auditory bulla (Figs. 2B and 4). A postglenoid foramen is present between the squamosal and periotic bones and is continuous with the posterior alar fissure (Figs. 2B, 2D, 2E and 4). The buccinator and masticatory foramina are fused and separate from the accessory foramen ovale, which is not within the alisphenoid (Figs. 2B, 2D and 5). The anterior squamosal foramen and stapedial foramen are both present, but the mastoid foramen is absent (Figs. 2B and 4). The posterior palatine foramen lies within the palatine-maxillary suture (Figs. 2B and 4). The foramen magnum is large and posteriorly oriented (Figs. 2B, 3C and 4). There are no vacuities situated anterior to the bullae or between the bullae and basioccipital, but a small fissure lies medial to the bullae (Fig. 4).

The rostrum is tapered anteriorly, both in the mediolateral and dorsoventral aspects (Figs. 2–5). The nasals descend anteriorly (Figs. 2D–2E, 3E and 5), and extend posteriorly past the premaxillae and anterior margin of the orbits (Figs. 2A and 4). The orbital constriction is just slightly broader than the posterior portion of the rostrum. The skull roof (frontals and parietals) is relatively flat, with prominent supraorbital ridges (Figs. 2A, 3A and 4). There are no clear postorbital processes and no laterally projecting supraorbital bony flange. The frontal projects anteriorly lateral to the posterior ends of the nasal and premaxilla, between the premaxilla and jugal (Figs. 2A, 3A and 4). The premaxilla frontal suture is highly interdigitated on the dorsal surface of the skull. The parietal is somewhat anteriorly retreated from the occiput, ending about mid-way anteroposteriorly along the auditory bulla (Figs. 2A, 3A and 4). The interparietal is wide but constricted by the inflation of the auditory bullae. The origin scar for the temporalis is shortened, lying well anterior to the occipital and restricted to the far lateral portion of the braincase. The infraorbital foramen is small and flattened, and a small but distinct tubercle lies along the posteroventral border of the foramen (Figs. 2B, 2D–2E and 5). The masseter is sciuromorphous, extending dorsally and anteriorly to the infraorbital foramen, along the rostral perforation (Figs. 2D–2E, 3E–3F). The area for the origin of the anterior zygomatic muscle is small and lies posterior to the infraorbital foramen (Fig. 3F). The squamosal is emarginate posteriorly, dorsal to the auditory bulla, and reduced to a thin bar at its most posterior extent (Figs. 2D, 2E and 5). The glenoid fossa lies anterodorsal to the auditory region; the jugal does not contribute to the glenoid fossa which is entirely within the squamosal (Figs. 2D, 2E and 5). A small boss lies anterior to the glenoid fossa, redirecting the temporalis muscle.

The posterior margin of the anterior root of the zygomatic arch lies lateral to the P4, anterior to the posterior end of the nasals and premaxillae (Figs. 2B, 3A–3B and 4). The basioccipital is greatly narrowed, and the basicranium is not swollen between the basioccipital and basisphenoid (Figs. 2B and 4). The alisphenoid does not extend far dorsally, has a narrow suture with the maxilla, and does not include any origin for the temporalis (Figs. 2E and 5). The upper tooth rows diverge posteriorly, and the main body of the palate extends posterior to the M3 (Figs. 2B, 3B and 4).

The upper incisor has a convex anterior face and lacks a central groove (Fig. 3D); it is broader anteroposteriorly than mediolaterally (Figs. 2B, 2D, 4 and 5). The upper cheek teeth are heavily worn (Figs. 2B–2E, 3 and 4), but some aspects of morphology are clear. The upper cheek teeth are mesodont and rooted, and enamel in the crown is continuous. P4 is larger than the molars, and the cheek teeth decrease in size posteriorly. The upper cheek teeth are bilophodont, with lophs fusing later in wear and forming a central enamel island when less worn (Fig. 2C). In the P4, the protoloph is fused to the metaloph medially, with the protoloph composed of a large and broad paracone that is fused along the posteromedial margin to the protocone. The P4 protocone merges laterally with the metacone, and the central valley between lophs opens anterolaterally. The M1 is heavily worn and clearly bilophodont; little other morphological detail is preserved. The M2s are also heavily worn with the lophs joined medially and laterally to form a central enamel island, which is open in the right M2 and nearly closed on the left M2. The M3 is less worn, and also has lophs joined both medially and laterally to form a central enamel island.

The dentary is nearly complete, anteroposteriorly elongate, and gracile (Figs. 2F, 2G, 3G–3I and 6). The diastema is anteroposteriorly elongate; the mental foramen lies ventral to the posterior end of the diastema, anterior to p4 and anteroventral to the masseter insertion (Figs. 2F, 3I and 6). The anterior end of the masseteric fossa extends anterior to p4, with the insertion marked by a strong anterior ridge (Figs. 2F and 3I). The posterior end of the incisor alveolus projects laterally from the dentary. A small foramen is present between the m3 and coronoid process (Figs. 2G, 3G, 6 and 7B). The mandibular condyle is anteroposteriorly oriented. The angle of the mandible is large and medially deflected (Figs. 2F–2G, 3G–3I and 6).

The lower incisor is characterized by a convex anterior face, is much broader anteroposteriorly than mediolaterally, has enamel extending onto ∼1/3 of the lateral surface, and is highly procumbent (149° angle) (Figs. 2F, 2G and 6). The lower p4 is slightly smaller than the m1 and the lower molars decrease in size posteriorly (Fig. 2H). The crowns of the lower cheek teeth are heavily worn, preventing the identification of individual cusps and lophs, but the teeth are clearly mesodont and rooted. The p4 is bilophodont, with the anterior loph narrower mediolaterally than the posterior loph. The lophs are joined medially and laterally, forming a distinct enamel island. The m1 is slightly broader mediolaterally than long anteroposteriorly, worn enough that lophs are not evident, and a has broad dentine filled basin surrounded by a ring of enamel. The m2 is also broader than long, clearly bilophodont, and has a small remnant of an enamel island in the center. The m3 is bilophodont, equal in length and breadth, and has lophs joined medially and laterally to form an enamel island.

The postcranial material preserved with the skull consists of a partial pes with relatively large and gracile bones (Figs. 2A and 4) and a caudal vertebra. The three incomplete metatarsals (missing proximal ends and with incomplete distal ends) lack sufficient preservation for detailed anatomical comparisons to those of other dipodomyines. The metatarsals have relatively oval cross sections that do not include flattened lateropalmar surfaces. The proximal phalanges are complete and relatively gracile. The caudal vertebra is large and robust (centrum W = 5.45 mm), consistent with anterior placement within the tail, but it is not preserved well enough for detailed comparisons or confirmation of its position.

Remarks.—UNSM 27016 has a suite of features that bear resemblance to extant and extinct dipodomyines and perognathines, but the combination of traits is unique and supports the designation of a new taxon. Figs. 4–6 highlight important morphological features that facilitate the comparison of this new taxon to two other important fossil heteromyids (Bursagnathus and Proheteromys) which have not been previously illustrated in such detail. The most prominent features of the skull are the inflated auditory bullae (Figs. 2–5). Although the ventral inflation of the bulla (#9, Table S5) varies across dipodomyines and is present in other taxa (e.g., the early perognathine Bursagnathus), the combination of a lateral inflation of the bulla (#10, Table S5) and the inflation of the dorsal, lateral, and posterior portions of the mastoid (#11–13, Table S5) is only seen in dipodomyines, including Aurimys (Figs. 4 and 5). The inflations of both the mastoid and bulla are also clearly evident in the micro-CT data, a sample coronal slice demonstrates the inflation of the mastoid and tympanic bones, as well as the single lamina bone of the latter (Fig. 7A). The buccinator and masticator foramina of Aurimys are fused (#23, Table S5, Fig. 5) as in Schizodontomys, Mioheteromys, Balantiomys, and Proheteromys, in contrast to derived dipodomyines, perognathines, and Bursagnathus where the foramina are separated. A distinct swelling is present at the posteroventral border of the infraorbital foramen (#28, Table S5, Fig. 5) in Aurimys as in Bursagnathus, Mioperognathus, Eochaetodipus, and Dipodomys, but unlike in other dipodomyines and perognathines.

Figure 6 Morphology of the occlusal and lateral views of the dentary of Aurimys xeros (UNSM 27016), Bursagnathus aterosseus (UCMP 56279), and Proheteromys latidens (UCMP 150688).

Selected anatomical features are highlighted, abbreviations are as follows: mt, mental foramen.

Figure 7 Selected anatomical traits of UNSM 27016, holotype specimen of Aurimys xeros, evident in micro-CT data.

(A)Coronal slice through the braincase and auditory region, with inflated mastoid and bulla evident, and single lamina bone visible, eam, external acoustic meatus (B) Coronal slice through the left dentary, with foramen between the coronoid process and m3 highlighted. Scale bars equal 1 cm and 1 mm, respectively. Skull CT image series: https://doi.org/10.17602/M2/M460741, Dentary CT image series: https://doi.org/10.17602/M2/M468758.

The rostrum of UNSM 27016 is tapered by anteriorly descending nasals (#48, Table S5), which is typical of many heteromyids, including dipodomyines, and contrasts with heteromyines, Chaetodipus, Mioheteromys, and Balantiomys, whose nasals do not descend. The posterior end of the nasals (#52, Table S5) of Aurimys extends farther posteriorly than the premaxillae, which is only seen in Bursagnathus among the studied heteromyids (Fig. 4). The suture between the premaxilla and frontal is interdigitated (#78, Table S5, Fig. 4), as in Schizodontomys, Bursagnathus, Mioperognathus, extant perognathines, Balantiomys, and Proheteromys, but unlike in other fossil and extant dipodomyines, which bear a simple suture. There is an anterior projection of the frontal between the premaxilla and jugal (#80, Table S5, Fig. 4) in UNSM 27016, which is also present in a wide range of taxa, including extant heteromyines, perognathines, and known dipodomyines; this projection is absent in Bursagnathus and Mioperognathus. The premaxillary-maxillary suture crosses the midline of the palate 1/3rd the distance from the posterior margin of the incisive foramen (#47, Table S5, Fig. 4) in UNSM 27016, where it is; that unusual character state is only observed elsewhere among the taxa studied in Mioheteromys, Balantiomys, Proheteromys, and Eochaetodipus. The suture crosses the midline at the posterior end of the incisive foramen in all other perognathines and at the midpoint of the incisive foramen in dipodomyines.

Table 2 Cranial measurements (in mm) of Aurimys xeros in comparison to other dipodomyines.

Measurement definitions appear in Table S2. For extant species, values represent species means, complete data for all specimens is provided in Table S3.

Species	Skull L	MCW	OrbW	Nasal L	Rost W	Rost D	Bulla L	Bulla W	RPL	RPW	UDiast L	ForMag W	Dent L	LDiast L	DentD m1	
Aurimys xeros UNSM 27016	48.50	26.50	10.52	17.54	9.90	11.65	17.21	9.13	5.26	3.31	15.51	6.70	34.81	9.01	6.64	
Prodipodomys mascallensis UCMP 39094														4.20	4.70	
Prodipodomys sp. AMNH 87427					7.35				2.40	1.37	8.84			4.22	3.85	
Dipodomys deserti (n = 2)	35.14	31.08	15.42	16.34	7.80	9.09	18.91	13.71	3.74	3.11	9.93	5.00	22.04	6.10	5.67	
Dipodomys heermani (n = 5)	30.70	24.23	11.79	13.87	6.87	8.41	11.20	9.62	3.32	2.67	9.36	4.79	19.23	5.01	4.50	
Dipodomys ingens (n = 5)	34.53	28.45	13.73	15.78	7.78	9.85	13.14	11.49	3.22	2.54	9.90	5.32	22.06	6.46	5.16	
Dipodomys merriami (n = 10)	27.45	22.59	12.88	12.52	5.82	7.39	14.17	9.81	2.92	2.43	7.83	4.10	16.73	3.79	4.19	
Dipodomys ordii (n = 1)	29.95	24.67	13.15		6.43	7.46	15.57	10.72	2.87	2.31	8.17	4.40	15.94	3.96	4.18	
Dipodomys spectabilis (n = 1)	34.52	29.49	17.01	15.87	8.19	9.80	17.58	13.58	3.18	3.02	11.33	5.36	23.91	6.17	6.30	
Microdipodops megacephalus (n = 5)	20.38	17.95	6.69	8.90	3.99	5.56	12.90	8.08	2.43	1.96	5.89	3.06	11.62	2.64	2.68	
Microdipodops pallidus (n = 5)	20.63	19.34	6.89	9.71	3.86	5.49	9.52	7.56	2.26	1.59	5.72	3.38	12.05	2.75	2.30	

The structure of the maxillary root of the zygomatic arch (#65 and 84, Table S5, Figs. 4 and 5) is distinctive in UNSM 27016. The posterior margin of the maxillary root lies lateral to P4, as in Bursagnathus, Cupidinimus, Eodipodomys, and Microdipodops; in Prodipodomys, Dipodomys, and extant perognathines it is slightly anterior to P4. Similar to Eochaetodipus, the posterior border of the maxillary root lies anterior to the posterior end of the nasals and premaxillae, in contrast to other perognathines and dipodomyines studied, where it is at or posterior to the level of the posterior end of the nasals and/or premaxillae. There is no supraorbital bony flange (#33, Table S5) present in UNSM 27016, like in some early heteromyids (Proheteromys, Mioheteromys, Balantiomys), but unlike the studied dipodomyines, perognathines, heteromyines, and Schizodontomys. The squamosal (#53, Table S5, Fig. 5) of UNSM 27016 is reduced to a thin bar posteriorly, as in extant perognathines and Schizodontomys, and in contrast to the unreduced squamosal of Bursagnathus, Mioperognathus, Eochaetodipus, and heteromyines, and the squamosal of extant dipodomyines which is overcome by the inflated mastoid and bulla. This aspect of the morphology of the squamosal is not known for the other fossil dipodomyines studied.

The parietal (#63, Table S5, Fig. 4) of UNSM 27016 is somewhat retreated from the occiput, as in Bursagnathus, Eochaetodipus, Mioperognathus, Schizodontomys, Cupidinimus, and extant heteromyines, and in contrast to the fully retreated parietal of Prodipodomys, extant dipodomyines, and extant perognathines. The interparietal (#31, Table S5, Fig. 4) of UNSM 27016 shows some constriction due to auditory bulla expansion, as in Eodipodomys and Cupidinimus, but not to the degree seen in more derived dipodomyines; this is also in contrast to Schizodontomys and perognathines, which lack the constriction of the interparietal by the auditory bullae. The origins of the temporal muscles (#64, Table S5) in UNSM 27016 are restricted very far laterally, as in extant perognathines and dipodomyines, and distinct from Mioheteromys, Balantiomys, Proheteromys, heteromyines, and Schizodontomys.

In UNSM 27016 the jugal does not contribute to the glenoid fossa (#79, Table S5, Fig. 5), which is also the case in known dipodomyines; in extant perognathines the jugal forms the anterolateral corner of the fossa and in heteromyines it forms the lateral wall of the fossa. The postglenoid foramen (#43, Table S5, Fig. 4) is present in UNSM 27016 as in Schizodontomys, Bursagnathus, and extant perognathines and heteromyines, in contrast with extant dipodomyines, Eodipodomys, and Mioperognathus. The foramen magnum (#70, Table S5, Fig. 4) of UNSM 27016 is posteriorly oriented, as is typical of all heteromyids other than extant dipodomyines, where it is anteriorly shifted and posteroventrally oriented. The position of the foramen magnum of Cupidinimus, Eodipodomys, and Prodipodomys is unknown.

The upper incisors of UNSM 27016 lack a central groove (#27, Table S5), similar to early dipodomyines and perognathines (Schizodontomys, Cupidinimus, Eodipodomys, Bursagnathus, and Eochaetodipus), but in contrast to later members of both groups (Prodipodomys, Dipodomys, Microdipodops, Chaetodipus, and Perognathus). The cheek teeth of UNSM 27016 lack chevrons (exposed dentine tracts at the base of the tooth crown, #86, Table S5, Figs. 2C and 2H), which is distinct from Cupidinimus, Eodipodomys, Prodipodomys, and extant dipodomyines, which all bear chevrons on the cheek teeth. Further, despite having heavy wear, the cheek teeth of UNSM 27016 have continuous enamel (#93, Table S5, Figs. 2C and 2H), a feature shared with perognathines, Cupidinimus, Prodipodomys, and Microdipodops, but distinct from Eodipodomys and Dipodomys. In the dentary of UNSM 27016, there is a small foramen between the m3 and coronoid process (#75, Table S5, Figs. 6 and 7B), which is also characteristic of Eodipodomys, Prodipodomys, and extant dipodomyines, and distinct from Cupidinimus, Bursagnathus, extant perognathines, extant heteromyines, Mioheteromys, Balantiomys, and Proheteromys where the foramen is absent.

Results of phylogenetic analysis

The parsimony analysis yielded 27 equally parsimonious trees. The strict consensus tree from the parsimony analysis (Fig. 8) and maximum clade credibility tree from the Bayesian analysis (Fig. 9) differ in a number of ways, but they also show a large number of similarities. Critically, both analyses place the new taxon, Aurimys xeros (UNSM 27016), as the earliest diverging taxon in a clade that includes all extant and fossil Dipodomyinae (Figs. 8 and 9). Both analyses also identify Prodipodomys as the sister taxon to Dipodomys with Microdipodops as sister to that clade. The subfamily Perognathinae (Perognathus, Chaetodipus) is a monophyletic group sister to Dipodomyinae in both analyses (Figs. 8 and 9). The taxa previously discussed as “stem perognathines”, Eochaetodipus, Mioperognathus, and Bursagnathus (Korth, 2008; Korth & Samuels, 2015), actually fall outside of Perognathinae + Dipodomyinae in both analyses (Figs. 8 and 9). The maximum parsimony analysis places the genus Schizodontomys as the sister group to the clade formed by the common ancestor of dipodomyines and Eochaetodipus whereas the position of the genus is unresolved at the base of crown group Heteromyidae in the Bayesian analysis. The Heteromyinae are the earliest diverging clade of Heteromyidae in the parsimony analysis. In both analyses, crown-group Heteromyidae consists of Heteromyinae, Schizodontomys, “stem perognathines”, Perognathinae, and Dipodomyinae. Relationships within the Geomyidae are identical in both the parsimony and Bayesian analyses (Figs. 8 and 9). The relationships of the families Heliscomyidae and Florentiamyidae to other geomorph taxa studied are similar in both analyses (Figs. 8 and 9). The relationships within each of those two families are also identical in the two analyses.

Figure 8 Consensus cladogram based on 27 most-parsimonious trees from parsimony analysis, derived from a matrix of 96 characters scored for 47 rodent taxa.

The matrix was 72.7% filled, TL:100.53, CI:0.478, RI:0.808, RCI:0.386. Well-recognized clades are highlighted by color.

Figure 9 Maximum clade credibility tree recovered from phylogenetic analysis using Bayesian inference, derived from a matrix of 96 characters scored for 45 geomorph rodent taxa using Paramys delicatus as the outgroup.

Numbers at nodes are posterior probability values. Well-recognized clades are highlighted by color.

The most obvious and important difference between the parsimony and Bayesian analyses is in the placement of the Geomyidae and taxa that have been considered basal heteromyids or stem geomyoids (Figs. 8 and 9). In the parsimony analysis, Geomyidae is the sister group to the crown-group Heteromyidae (Fig. 8), and several taxa previously considered to be ‘basal’ heteromyids by Flynn, Lindsay, & Martin (2008: 436), sometimes included in the clade ‘Mioheteromyinae’ (Korth, 1997), like Proheteromys, Mioheteromys, Balantiomys, and Trogomys, are placed as stem geomyoids along with taxa considered as such by Flynn, Lindsay & Martin (2008), like Proharrymys, Harrymys, and Tenudomys as well as the Mojavemyinae (Mojavemys + Phelosaccomys). In contrast, in the Bayesian analysis, those taxa (Proharrymys, Harrymys, Tenudomys, mojavemyines, and ‘mioheteromyines’) are all early diverging members of the Heteromyidae, with Geomyidae as sister to that larger group (Fig. 9).

There are some unresolved polytomies evident in both the parsimony and Bayesian analyses. Thus, there is an unresolved polytomy of Sanctimus species in both analyses (Figs. 8 and 9), and Schizodontomys species in the Bayesian analysis (Fig. 8). Additionally, the relationships among Proheteromys, Balantiomys, Mioheteromys, and Mojavemys + Phelosaccomys, their relationship to Heteromyidae, and the relationships of Tenudomys, Harrymys, and Proharrymys are unresolved in the Bayesian analysis (Fig. 9).

Results of analyses of body size and crown height evolution

The cranial, dental, and postcranial dimensions of Aurimys xeros (UNSM 27016) were compared with both modern and fossil dipodomyine specimens (Tables 2–4, Table S3). Across cranial and dental measurements, Aurimys is consistently larger than any extant heteromyid (Table 2, Table S3), with the exceptions of maximum cranial width (MCW), orbital width (OrbW), and auditory bulla length and width (BullaL, BullaW). The skull length and cheek tooth row lengths (P4-M3L, p4-m3L) of Aurimys are over 1/3rd larger than the studied specimens of Dipodomys (Tables 2–3, Table S3), including specimens of the largest known extant heteromyids, D. ingens and D. spectabilis (Williams, Genoways & Braun, 1993; Brylski, 1993; Noftz & Calede, 2022). The auditory bullae of Aurimys are large, but do not show an inflation as extreme as in Dipodomys, therefore they are smaller than in the largest members of that genus and also proportionately smaller than in either Dipodomys or Microdipodops (Table 2, Table S3). In dipodomyines, maximum cranial width is directly related to the size of the auditory bullae, therefore the maximum cranial width (MCW) of Aurimys is smaller than in the largest extant species of Dipodomys (Table 2, Table S3).

Table 3 Dental measurements (in mm) of Aurimys xeros and other dipodomyines.

Measurement definitions appear in Table S2. For extant species, values represent species means, complete data for all specimens is provided in Table S3. Mean values for Prodipodomys idahoensis from Czaplewski (1990).

Species	Specimens		I1L	I1W	P4L	P4W	M1L	M1W	M2L	M2W	M3L	M3W	P4-M3L	
Aurimys xeros	UNSM 27016	Left	2.66	1.62	2.42	2.47	1.92	2.63	1.84	2.46			9.41	
		Right			2.39	2.49	1.87	2.56	1.80	2.35	1.66	1.93	9.38	
Cupidinimus nebraskensis	CMNH 10193				0.86	1.05					0.58	0.81		
	CMNH 10170				1.09	1.12	0.90	1.08	0.80	0.96	0.67	0.80	3.20	
Eodipodomys celtiervator	UGV 109				2.10	2.00	1.45	2.30						
Prodipodomys idahoensis	Czaplewski (1990)				1.29	1.68	1.07	1.67	0.96	1.44	0.77	0.90	5.02	
Prodipodomys sp.	AMNH 87427				1.47	0.92	1.25	1.41	1.15	1.52				
Dipodomys deserti	various (n = 2)		1.90	1.13	1.29	2.01	1.21	2.14	1.21	2.08	1.14	1.40	5.11	
Dipodomys heermanni	various (n = 5)		1.52	1.07	1.05	1.47	1.02	1.60	0.97	1.42	0.94	1.06	4.86	
Dipodomys ingens	various (n = 5)		1.88	1.32	1.25	1.63	1.22	1.83	1.10	1.64	1.11	1.29	5.10	
Dipodomys merriami	various (n = 10)		1.64	0.85	1.05	1.54	0.90	1.65	0.86	1.50	0.83	1.18	3.83	
Dipodomys ordii	various (n = 1)		1.60	0.99	1.10	1.74	0.93	1.73	0.90	1.50	0.82	1.12	3.93	
Dipodomys spectabilis	various (n = 1)		2.04	1.40	1.20	2.02	1.29	2.23	1.12	1.94	1.01	1.55	4.70	
Microdipodops megacephalus	various (n = 5)		1.18	0.62	0.97	1.12	0.72	1.12	0.64	0.89	0.55	0.67	3.00	
Microdipodops pallidus	various (n = 5)		1.07	0.65	1.06	1.11	0.77	1.15	0.65	0.88	0.52	0.56	3.42	
			i1L	i1W	p4L	p4W	m1L	m1W	m2L	m2W	m3L	m3W	p4-m3L	
Aurimys xeros	UNSM 270166	Left	2.52	1.63	2.04	2.01	1.86	2.37	1.86	2.56	1.91	2.08	8.58	
Cupidinimus nebraskensis	CMNH 10193				0.84	0.89	0.87	1.02	0.77	0.91	0.57	0.69	3.01	
Eodipodomys celtiervator	UGV 109		2.40	1.25	2.70	1.80	1.95	2.00	1.40	1.85	0.80	1.15	6.55	
Prodipodomys idahoensis	Czaplewski (1990)				1.30	1.35	1.10	1.57	0.95	1.46	0.73	1.09	5.19	
Prodipodomys mascallensis	UCMP 39094		1.50	0.80	1.00	1.20	1.40	1.40	1.30	1.50	1.20	1.30	4.80	
Prodipodomys sp.	AMNH 87427		1.29	0.73	1.17	1.27	1.18	1.42	1.03	1.42	0.81	0.99	4.79	
Dipodomys deserti	various (n = 2)		1.64	0.94	1.35	1.70	0.96	1.83	1.02	1.65	0.85	1.31	5.36	
Dipodomys heermanni	various (n = 5)		1.37	0.87	1.31	1.39	1.00	1.54	0.95	1.39	0.76	1.07	4.32	
Dipodomys ingens	various (n = 5)		1.59	1.00	1.50	1.54	1.14	1.78	1.05	1.65	0.87	1.24	5.19	
Dipodomys merriami	various (n = 10)		1.35	0.71	1.19	1.39	0.86	1.48	0.81	1.43	0.71	1.10	3.83	
Dipodomys ordii	various (n = 1)		1.18	0.74	1.14	1.57	0.84	1.62	0.82	1.53	0.74	1.12	3.86	
Dipodomys spectabilis	various (n = 1)		1.76	1.23	1.68	1.77	1.18	2.19	0.95	2.00	0.91	1.68	4.54	
Microdipodops megacephalus	various (n = 5)		0.95	0.58	1.07	0.97	0.72	1.08	0.68	1.05	0.51	0.73	3.16	
Microdipodops pallidus	various (n = 5)		0.96	0.51	0.95	0.99	0.79	1.07	0.67	0.95	0.56	0.65	3.26	

Table 4 Pedal measurements (in mm) of Aurimys xeros and other dipodomyines, as well as Schizodontomys harkseni (from Munthe, 1981).

Measurement definitions appear in Table S2. For extant species, values represent species means, complete data for all specimens is provided in Table S3.

Species	Specimens	mt3W	mt4W	pph3L	pph3W	pph4L	pph4W	
Aurimys xeros	UNSM 27016	1.91	1.99	8.07	1.54	6.57	1.13	
Cupidinimus nebraskensis	CMNH 10193	0.80	0.70					
Prodipodomys sp.	AMNH 87427	1.19	0.95	5.38	0.81	5.52	0.77	
Dipodomys deserti	various (n = 3)	1.48	1.33	8.73	1.15	8.46	1.17	
Dipodomys heermanni	ETMNH Z17752			9.28	1.10	9.12	1.09	
Dipodomys merriami	various (n = 3)	1.07	0.99	5.94	0.79	5.67	0.74	
Dipodomys spectabilis	ETMNH Z2255	1.38	1.31	6.62	1.20	6.43	1.21	
Dipodomys sp.	ETMNH Z10311			7.63	1.22	7.40	1.19	
Schizodontomys harkseni	UCMP 113568	2.6	2.44					

Among fossil dipodomyines represented by relatively complete material, there is a wide range of sizes represented. Aurimys xeros has teeth larger than the largest extant heteromyids studied (Dipodomys deserti, D. ingens, D. spectabilis) and Eodipodomys celtiservator has teeth comparable in size to those taxa (Table 3, Table S3). Prodipodomys specimens fall within the size ranges of most species of extant kangaroo rats (Dipodomys) (Table 3, Table S3). In contrast, Cupidinimus nebraskensis specimens are comparable in size to the smallest extant dipodomyines, the kangaroo mice (Microdipodops) (Table 3, Table S3).

Although limited to a few elements, pedal measurements of Aurimys xeros were compared to several extant kangaroo rats and fossils of Cupidinimus nebraskensis, Prodipodomys sp., and Schizodontomys harkseni. The mediolateral width of the 3rd and 4th metatarsals (mt3W, mt4W) of Aurimys were greater than in any studied specimens of Dipodomys, but smaller than those of S. harkseni (Table 4, Table S3). Lengths of the proximal phalanges of digits 3 and 4 (pph3L, pph4L) and width of the proximal phalanx of digit 4 (pph4W) of Aurimys fall within the range of variation of extant Dipodomys specimens, but the width of the proximal phalanx of digit 3 is greater than any extant or fossil specimen studied (Table 4, Table S3).

The cheek tooth crown heights of crown-group Heteromyidae were examined in a phylogenetic framework. The values for individual taxa and reconstructed ancestral character states mapped onto the time-scaled consensus cladogram from the parsimony analysis is shown in Fig. 10. Mesodont crown height is reconstructed as the ancestral state of crown-group heteromyids, and all early Miocene (Arikareean and Hemingfordian) taxa have mesodont crown height with the exception of Eochaetodipus asulcatus, which is brachydont (Fig. 10). The subfamilies Heteromyinae and Perognathinae are reconstructed as maintaining mesodont crown height through the late Cenozoic (except for Mioperognathus), but the subfamily Dipodomyinae displays a pattern of increased crown height from the Miocene on (Fig. 10). The analysis suggests hypsodont crown height evolved in Dipodomyinae in the early Miocene, either in the late Arikareean or early Hemingfordian and hypselodont crown height appeared sometime after the late Miocene (Fig. 10).

Figure 10 Crown height evolution of crown-group heteromyids.

Phylogenetic relationships based on consensus cladogram from parsimony analysis (Fig. 8). Geologic ages of known fossils are represented by thick black lines, and derived from Samuels & Hopkins (2017). Divergence estimates for extant clades are based on Hafner et al. (2007). Character state evolution of crown height is shown, with squares adjacent to names indicating character states at the tips and rectangles at nodes showing reconstructed ancestral character states. Colors indicate crown height values: yellow, brachydont; orange, mesodont; red-orange, hypsodont; dark red, hypselodont. 50% marginal likelihood is represented by tick mark to the right of the rectangle.

Discussion

The new dipodomyine described here, Aurimys xeros, represents the 57th rodent species known from the John Day Formation (Korth & Samuels, 2015; Samuels & Korth, 2017), and one of only a few described from the Early Miocene aged Johnson Canyon Member. The morphology of the new taxon is unlike any previously known heteromyid, showing a mosaic of ancestral heteromyid traits and highly derived features characteristic of Dipodomyinae. Phylogenetic analyses support the placement of Aurimys within the Dipodomyinae, making it the earliest known member of the kangaroo rat/mouse clade. Dating to the Early Miocene (late or latest Arikareean) and in close stratigraphic proximity to a 22.746 ± 0.146 radioisotopically-dated tuff, the new species is potentially several million years older than any other dipodomyine. Prior to this, the earliest published record of a dipodomyine was Cupidinimus boronensis from the early Hemingfordian-aged Boron Local Fauna and late Hemingfordian-aged Vedder Local Fauna of California (Whistler, 1984; Whistler & Lander, 2003). The discovery of Aurimys xeros therefore provides the unique opportunity to investigate the origin of dipodomyines, the ancestral ecology of the clade, and how they evolved and spread across North America.

Phylogeny of heteromyidae and geomorpha

The phylogenetic analyses employed here include the broadest sampling of living and extinct geomorph rodents and characters of any morphological study to date. The only other cladistic analysis of Geomorpha previously published (Wahlert, 1991) used a matrix more than eight times smaller than the one analyzed in this study. These results reveal the relationships of the new fossil taxon to known clades and help resolve the relationships of taxa within Heteromyidae and more broadly Geomorpha. Thus, results demonstrate the existence of a monophyletic crown-group Heteromyidae exclusive of Geomyidae in both parsimony and Bayesian analyses (Figs. 8 and 9). Aurimys xeros (UNSM 27016) is consistently recovered as the earliest diverging taxon in a clade that includes all extant and fossil Dipodomyinae (Figs. 8 and 9). The sister group to dipodomyines in both analyses is the Perognathinae. This contrasts with the relationships among heteromyid subfamilies recovered in recent molecular studies, which consistently place Heteromyinae and Perognathinae as sister clades and Dipodomyinae outside of that group (Hafner et al., 2007; Fabre et al., 2012; Upham, Esselstyn & Jetz, 2019). Wahlert (1991) also recovered Perognathinae as sister to Dipodomyinae using a different matrix of morphological characters. These contrasting morphological and molecular findings suggest that similar cranial structures (like inflated auditory bullae) and dental features (central fusion of lophs in the p4) may have arisen independently and evolved in parallel through time in perognathines and dipodomyines. Eochaetodipus, Mioperognathus, and Bursagnathus were previously discussed as “stem perognathines” by Korth (2008) and Korth & Samuels (2015), but they are found to be outside of Perognathinae + Dipodomyinae in both parsimony and Bayesian analyses here (Figs. 8 and 9). The phylogenetic relationships of Schizodontomys have long been difficult to identify (Korth, Bailey & Hunt Jr, 1990; Wahlert, 1993) with some older studies even placing the genus within Geomyidae (Rensberger, 1973a). Results in this study demonstrate that it falls towards the base of crown-group Heteromyidae, a position in contrast to previous suggestions that (1) it is a member of a monophyletic (Korth, 1997) or paraphyletic (Flynn, Lindsay & Martin, 2008) ‘Mioheteromyinae,’ a purported clade for which this study does not currently recover any evidence, (2) it is an early member of Dipodomyinae (Wahlert, 1985), or (3) it is a member of Heteromyinae (Korth, Bailey & Hunt Jr, 1990). Within crown-group Heteromyidae, Heteromyinae are recovered as the earliest diverging clade in the parsimony analysis (Fig. 8) whereas the exact topology remains unresolved in the Bayesian analysis (Fig. 9).

Analyses yield different topologies for the placement of Geomyidae (Figs. 8 and 9). The parsimony analysis places Geomyidae as the sister group to crown-group Heteromyidae (Fig. 8), but the Bayesian analysis has several taxa previously considered basal heteromyids and stem geomyoids as more closely related to crown-group Heteromyidae than Geomyidae (Fig. 9). Other molecular and morphological studies also vary in the placement of Geomyidae, either having Geomyidae outside of a monophyletic Heteromyidae (Hafner et al., 2007; Upham, Esselstyn & Jetz, 2019) or having Heteromyidae as a paraphyletic group with Geomyidae nested within (DeBry, 2003; Fabre et al., 2012; Asher et al., 2019; Upham, Esselstyn & Jetz, 2019). None of those analyses included a broad sampling of fossil geomorphs, however, and results of this study are consistent with those of Hafner et al. (2007) and Upham, Esselstyn & Jetz (2019). The exact placement of the node Heteromyidae in a systematic framework including fossils is critical to determining if Geomyidae is nested within Heteromyidae. Harrymys was previously placed within Heteromyidae by Wahlert (1991) and as a stem geomyoid by Flynn, Lindsay & Martin (2008); the position of Harrymys and other similar taxa relative to Heteromyidae and Geomyidae vary across analyses in this study, nested near Heteromyidae in the Bayesian analysis (Fig. 9) and as stem geomyoids in the parsimony analysis (Fig. 8).

Although the placement of Geomyidae within Geomorpha is different in the parsimony and Bayesian analyses, the family is monophyletic and the relationships of taxa within each subfamily are identical in both (Figs. 8 and 9). The Entoptychinae and Geomyinae are sister taxa (Figs. 8 and 9), as was also found by Wahlert (1991). This is in contrast with Asher et al. (2019), which had a polyphyletic Geomyidae placing the entoptychine Gregorymys outside of Geomyoidea; this is likely a consequence of taxonomic and character sampling. Analyses recovered the families Heliscomyidae and Florentiamyidae as early diverging groups of geomorph rodents outside of the Geomyoidea (Figs. 8 and 9), corroborating the hypothesis of Flynn, Lindsay & Martin (2008) and, for Florentiamyidae, the findings of Wahlert (1991). This finding also contrasts with Asher et al. (2019) who had the florentiamyid Florentiamys placed with the entoptychine geomyid Gregorymys as sister to the eomyid Paradjidaumo outside of the Geomyoidea, which they interpreted as a group that included Heliscomys.

The addition of more characters to the phylogenetic analyses will be critical to help resolve some polytomies, particularly the polytomies within Sanctimus and Schizodontomys. Adding select dental characters could also offer some resolution and further support, but the utility of those traits may be limited given the convergence in dental morphology and retention of ancestral traits within geomyoids (Wahlert, 1993). Future analyses would benefit from the additional confidence provided by a bootstrap or Bremer decay indices, which were not computationally possible here. Results of these phylogenetic analyses offer the opportunity to study the evolution of Heteromyidae (and more broadly Geomorpha), including possible convergence of cranial traits and features of the dentition like the x-pattern of the p4 that characterizes Dipodomyinae and Perognathinae.

Paleoecology of early dipodomyines

The well-preserved holotype of Aurimys xeros (UNSM 27016) provides important evidence of the ecology of early dipodomyines, based on body size, craniodental structure, and some postcranial features. Aurimys xeros is the largest known heteromyid (Tables 2–4, Table S3); it is approximately 1/3rd larger than the largest living heteromyids, all species of kangaroo rats in the genus Dipodomys (Best, 1993; Brylski, 1993; Williams, Genoways & Braun, 1993; Noftz & Calede, 2022). Next to A. xeros, the largest known fossil heteromyids are species of Schizodontomys (Munthe, 1981). The next largest known fossil dipodomyine is Eodipodomys celtiservator (Voorhies, 1975) which is comparable in size to the large extant kangaroo rats D. deserti, D. ingens, and D. spectabilis (Table 3, Table S3). The size of Eodipodomys contrasts strongly with the contemporaneous Cupidinimus, which is similar in size to the smallest extant dipodomyines, the kangaroo mice (Microdipodops) (Table 3, Table S3). These records suggest a complex pattern of body size evolution within Heteromyidae, with large body size evolving possibly four times (in Schizodontomys, Aurimys, Eodipodomys, and Dipodomys). Alternatively, large body size may have been a characteristic of the earliest dipodomyines, which was subsequently lost in some species of Dipodomys and Prodipodomys as well as Cupidinimus and Microdipodops. Either way, a large body size disparity was present in crown-group Heteromyidae in the early and middle Miocene, with taxa occupying similar size niches to extant dipodomyines through the late Neogene and Quaternary. Future phylogenetic comparative analyses including additional taxa will be necessary to rigorously test these hypotheses of body size evolution.

The skull, dentary, and dental morphology of Aurimys can also be directly compared to extant rodents with known dietary habits, including many well-studied heteromyids. The cheek teeth of the Aurimys specimen are worn, but their lophodont occlusal morphology (Figs. 2C and 2H) and mesodont crown height (Fig. 10) are similar to many extant and fossil heteromyids (Brylski, 1993; Wahlert, 1993; Flynn, Lindsay & Martin, 2008; Samuels & Hopkins, 2017). The same is true of the anteroposteriorly deep and mediolaterally broad incisors of Aurimys (Figs. 2B and 2D, 2F, 2G, 4, 5 and 6, Table 3, Table S3). The cranial morphology of Aurimys shows similarity to extant dipodomyines and perognathines, displaying relatively gracile zygomatic arches, lacking prominent sagittal crests, and lacking increased skull depth (Figs. 2A, 2D, 2E, 3, 4 and 5). All of these craniodental features are similar to those of omnivore and generalist herbivore rodents from multiple families (Samuels, 2009) and suggest Aurimys was a plant-dominated omnivore like extant dipodomyines and perognathines (Reichman & Price, 1993). Future examination of the procumbency, bite force, and enamel microstructure of Aurimys and other fossil heteromyids could offer additional insights into their ancestral ecology (Kalthoff & Mörs, 2021).

The auditory bullae of Aurimys xeros are large, showing a pronounced inflation of both the tympanic and mastoid bones (Figs. 2, 4, 5 and 7A). The combination of anterior, ventral, and lateral inflation of the tympanic and dorsal, lateral, and posterior portions of the mastoid is characteristic of dipodomyines and distinct from other heteromyids (Best, 1993; Lay, Genoways & Brown, 1993; Data S1). The inflation of the auditory region in A. xeros is not as extreme as the inflation observed in Dipodomys or Microdipodops (Table 2, Table S3). Inflated auditory bullae have been extensively studied across rodent clades (Webster, 1962; Webster & Webster, 1971; Webster & Webster, 1975; Webster & Webster, 1984; Lay, Genoways & Brown, 1993; Alhajeri, Hunt & Steppan, 2015; Alhajeri & Steppan, 2018; Scarpitti & Calede, 2022). Evidence suggests that hypertrophied bullae represent an adaptation to sound amplification and predator avoidance in open habitats (Alhajeri, Hunt & Steppan, 2015; Alhajeri & Steppan, 2018), although it is important to note that there is no direct association between the auditory bulla size of heteromyids and the aridity of the environment they inhabit (Webster & Webster, 1975; Scarpitti & Calede, 2022).

The type specimen of Aurimys xeros preserves only a few partial postcranial elements, but there is sufficient cranial and postcranial material to interpret its locomotor habits. The foramen magnum of Aurimys is oriented posteriorly (Figs. 2B and 4), as in all other crown-group heteromyids other than the extant dipodomyines Dipodomys and Microdipodops (note that other fossil dipodomyines do not have a preserved foramen magnum). That posterior orientation is typical of quadrupedal rather than bipedal rodents (Russo & Kirk, 2013; Russo & Kirk, 2017). The partial pes of Aurimys is large but not preserved well-enough for a comparison of limb proportions (as in Samuels & Van Valkenburgh, 2008; Calede, Samuels & Chen, 2019), and the pedal phalanges are smaller compared to the size of the skull than in extant dipodomyines (Table 4). In contrast to extant dipodomyines and Eodipodomys (Voorhies, 1975), the metatarsals of Aurimys do not have flattened lateropalmar surfaces and likely had more interdigital mobility than is typical of ricochetal rodents. Despite having the largest size of any known heteromyid, the preserved metatarsals of A. xeros are more gracile than those of Schizodontomys (Table 4). The only caudal vertebra of Aurimys is large and elongate, and bears clear similarity to the vertebra described for Cupidinimus (Wood, 1935). Aurimys is interpreted as a quadrupedal saltator, similar to the habits Munthe (1981) suggested for Schizodontomys harkseni and the habits typical of many extant heteromyids (Hafner, 1993).

Quadrupedal saltatory locomotion has been hypothesized to be ancestral for geomyoids (Gambaryan, 1974; Brylski, 1993; Hafner, 1993; Scarpitti & Calede, 2022). The morphology of known fossil taxa and phylogenetic results are consistent with that interpretation. Although quadrupedal saltation (hopping) is common among rodents, primarily ricochetal (bipedal) locomotion is restricted among geomorph rodents to dipodomyines (Hatt, 1932; Howell, 1932; Wood, 1935; Gambaryan, 1974; Brylski, 1993; Samuels & Van Valkenburgh, 2008). The characteristics of Aurimys and the phylogenetic results of this study suggest that the Dipodomyinae were ancestrally large bodied quadrupedal saltators, with bipedality and ricochetal locomotion arising sometime prior to the early late Miocene diversification of crown-group dipodomyines. The study of a Pliocene specimen of Prodipodomys sp. (AMNH F:AM 87427) supports this interpretation. Indeed, the morphology and dimensions of the pes of Prodipodomys fall within the range of extant Dipodomys (Table 4). The interpretation of geomyoid locomotor evolution in this study contrasts with that of Asher et al. (2019), who described a skeleton of Heliscomys ostranderi from the latest Eocene of Wyoming and included it in a phylogenetic analysis of rodents. They interpret H. ostranderi as having been ricochetal, based on somewhat elongate metatarsals and slightly reduced peripheral digits. Based on their phylogenetic analysis, they interpret ricochetal locomotion as having been ancestral in geomyoids (Asher et al., 2019). However, H. ostranderi has a pedal morphology similar to that of quadrupedal saltators within Heteromyidae like Heteromys and Chaetodipus and lacks the limb proportions characteristic of ricochetal/bipedal rodents, including dipodomyines (Howell, 1932; Samuels & Van Valkenburgh, 2008; Moore et al., 2015; Calede, Samuels & Chen, 2019). The orientation of the foramen magnum of Heliscomys (posteriorly oriented, Data S1) is also consistent with a quadrupedal rather than bipedal locomotion (Russo & Kirk, 2013; Russo & Kirk, 2017).

The Johnson Canyon environment and fauna

Paleosol records from the late Oligocene and Early Miocene of Oregon document a transition to cooler, drier conditions (Sheldon, Retallack & Tanaka, 2002; Retallack, 2004; Retallack, 2007). Faunal evidence also supports the interpretation that habitats of the John Day Basin were changing in the Early Miocene, with open habitat adapted species becoming more common (Hunt Jr & Stepleton, 2004; Samuels & Schap, in press). The Johnson Canyon fauna records several taxa that have been interpreted as adapted for open habitats, including the early mylagaulid Mylagaulodon, the heteromyids Schizodontomys and Bursagnathus, the leporid Archaeolagus, the amphicyonid Temnocyon, the equid Kalobatippus, and the camelid Paratylopus (Hunt Jr & Stepleton, 2004; Calede, Hopkins & Davis, 2011; Hunt Jr, 2011; Korth & Samuels, 2015; Samuels & Kraatz, 2015; Samuels & Hopkins, 2017). Archaeolagus, Paratylopus, Kalobattipus, and Temnocyon all occur in earlier John Day strata (Hunt Jr & Stepleton, 2004; Fremd, 2010), but the co-occurrence of these taxa, all interpreted as cursorially-adapted and of a wide range of body sizes, suggests that relatively open habitats were prevalent in Oregon at the time. Channel conglomerates from the Johnson Canyon strata contain abundant silicified wood alongside animal remains, indicating the presence of wooded riparian habitats, which would be home to the small chalicothere Moropus, the tapir Miotapirus, and browsing equids (anchitheres, Archaeohippus) known from the assemblage (Hunt Jr & Stepleton, 2004). The fauna also includes early North American occurrences of cervoids (the moschid Pseudoblastomeryx and an unidentified dromomerycid) and a lower diversity of oreodonts than older John Day Formation strata (Hunt Jr & Stepleton, 2004; Fremd, 2010), documenting important changes in the mammalian herbivore community. Although only few carnivorans are known from the Johnson Canyon fauna, there are two potential predators of Aurimys, the borophagine canids Desmocyon and Cynarctoides (Hunt Jr & Stepleton, 2004).

As discussed above, the inflation of the auditory bullae is considered an adaptation for auditory acuity and arid environments. Inflated auditory bullae are evident in two rodents from the Johnson Canyon fauna, both heteromyids, Aurimys and Bursagnathus (Korth & Samuels, 2015). The facts that these two pericontemporaneous taxa demonstrate auditory bulla inflation and are two of the earliest heteromyids to show that trait, suggest they show adaptations to cooler and more arid environmental conditions in Oregon during the Early Miocene (Sheldon, Retallack & Tanaka, 2002; Retallack, 2004; Retallack, 2007). Interestingly, one of the potential predators of those heteromyids, Cynarctoides, is known to bear enlarged auditory bullae (Wang, Tedford & Taylor, 1999). Cynarctoides lemur from the Turtle Cove and Kimberly Members and C. cf. luskensis from the Johnson Canyon Member both have relatively large auditory bullae, possibly indicating adaptation for improved hearing abilities among some small predators at the time.

The evolution and spread of dipodomyines

The species described here represents the earliest appearance of dipodomyine heteromyids. Other early records of dipodomyines are from the early Hemingfordian (early Miocene) of the California Coast and Great Basin (Whistler, 1984; Whistler & Lander, 2003; Samuels & Schap, in press). Dipodomyines spread to the Northern Rocky Mountains in the early Barstovian (middle Miocene) and Northern Great Plains in the late Barstovian (Samuels & Schap, in press). The dipodomyines finally spread to the Colorado Plateau in the early late Hemphillian (late Miocene), Southern Great Plains in the latest Hemphillian (early Pliocene), and Southwest in the early Blancan (early Pliocene) (Samuels & Schap, in press).

Samuels & Schap (in press) hypothesized that the Columbia Plateau may have been a cradle for rodent evolution through the Cenozoic. The region was tectonically active and topographically complex (Badgley, 2010; Mix et al., 2011; Kent-Corson et al., 2013; Badgley et al., 2017; Kukla et al., 2021) and had pronounced volcanic activity across much of the Cenozoic (McClaughry et al., 2009). The Columbia Plateau is at relatively high latitude, has altitudinal variation, and in the past had potential rain shadow effects (Kohn & Fremd, 2007; Kohn & Fremd, 2008; Retallack, 2007; Retallack, 2009; Chamberlain et al., 2012; Kukla et al., 2021), which may have yielded relatively cool and arid conditions (Graham, 1999; Pound et al., 2012; Pound & Salzmann, 2017; Schap, Samuels & Joyner, 2021). Paleobotanical evidence and paleoclimate records from paleosols suggest that the aridification and opening of habitats in Oregon began during the Oligocene (Retallack, 2007; Retallack, 2009; Dillhoff et al., 2009). As aridification continued through the late Oligocene and early Miocene, there was a clear transition to grass-dominated habitats including open woodlands and scrublands (Strömberg, 2011; Kukla et al., 2022). The pattern of crown height evolution in heteromyids reveals ancestrally mesodont cheek teeth in the early history of the family (and early dipodomyines), and increased crown height through the Miocene (Fig. 10), coincident with the aridification of the region and the onset of the open habitat transition in North America (Kukla et al., 2022). The relatively cool and arid conditions of the early Miocene (Westerhold et al., 2020), and the prevalence of grit (from aridification and volcanic ash) could have driven increased tooth crown height in the group through time (Fig. 10). In addition to increased cheek tooth crown height, other characteristic features of dipodomyines, specifically predator evasion adaptations like auditory bullae inflation and saltatory locomotion, may have been driven by the expansion of open habitats. As open habitats spread through the Miocene, dipodomyines (adapted to such conditions) thrived, eventually becoming one of the most species-rich and abundant groups of mammals in western North America (Kotler & Brown, 1988; Genoways & Brown, 1993; Hafner, 1993; Hafner et al., 2007; Flynn, Lindsay & Martin, 2008; Samuels & Hopkins, 2017).

Auditory and locomotory specializations facilitate predator avoidance in open habitats (Bartholomew & Caswell, 1951; Eisenberg, 1963; Kotler, 1985; Djawdan & Garland Jr, 1988; Longland & Price, 1991; Freymiller et al., 2019). Two of the most important predators of extant heteromyids are owls (Kotler, 1985; Brown et al., 1988; Longland & Price, 1991; Terry, 2010) and rattlesnakes (Beavers, 1976; Pierce, Longland & Jenkins, 1992). Owls have a long history in North America (Rich & Bohaska, 1976; Rich & Bohaska, 1981; Mayr, Gingerich & Smith, 2020) and were certainly present as heteromyids diversified over time, with the extant family Strigidae possibly first represented by Strix dakota from the early Miocene (Hemingfordian) of Nebraska (Miller, 1944). Interestingly, the earliest record of a viperid in North America is also from the early Miocene (late Arikareean) (Holman, 2000). The evolution of dipodomyines and their specializations may be a product of coevolution with owls and rattlesnakes through the Neogene.

Conclusions

The new genus and species of dipodomyine heteromyid from the early Miocene of Oregon described herein represents the earliest known record of the kangaroo rat/mouse clade. Aurimys xeros has a mosaic of morphological features, including some dental and cranial characteristics like those of early heteromyids along with highly inflated auditory bullae, a derived adaptation of dipodomyines. Phylogenetic analysis of living and extinct geomorph rodents used here represent the broadest sampling of both taxa and morphological traits of any study to date. Findings support the monophyly of crown-group Heteromyidae and place Aurimys xeros within Dipodomyinae. The new taxon is the largest known heteromyid, and analyses suggest large body size evolved several times within the family. The morphology of the new taxon and other fossil dipodomyines reveal a mosaic evolution of open-habitat adaptations in kangaroo rats and mice, with the inflation of the auditory bulla showing up early and bipedality/ricochetal locomotion appearing later in the history of the group. The appearance of open habitat adaptations corresponds with cooling and drying conditions in the late Oligocene and early Miocene, and the specialization of dipodomyines over time was likely driven by late Cenozoic climate and habitat changes in North America.

Supplemental Information

Table S1 Modern and fossil specimens included in the phylogenetic analyses of this study

Click here for additional data file.

Table S2 Definitions of cranial, dental, and postcranial measurements included in this study

Dental measurements follow Carrasco (2000), cranial measurements follow Korth & Samuels (2015), and postcranial measurements follow Samuels & Van Valkenburgh (2008).

Click here for additional data file.

Table S3 Complete cranial, dental, and postcranial measurements of studied specimens

Definitions of measurements are provided in Table S2.

Click here for additional data file.

Table S4 Character list and character states for phylogenetic analyses

A complete matrix of character states for all studied taxa is provided in Table S5, while data files used in parsimony and Bayesian analyses are provided in Data S1 and S2, respectively.

Click here for additional data file.

Table S5 Complete matrix of character states for all studied taxa used for phylogenetic analyses

Data files used in parsimony and Bayesian analyses are provided in Data S1 and S2, respectively.

Click here for additional data file.

Data S1 Nexus file used in parsimony phylogenetic analyses, including character states for all studied taxa

Click here for additional data file.

Data S2 Nexus file used in Bayesian phylogenetic analyses, including character states for all studied taxa

Click here for additional data file.

UNSM 27016 was found by and expertly prepared by Ellen Stepleton, her observations and care have helped make this project possible. We thank the following people for access to specimen collections: Kathy Molina (UCLA), Pat Holroyd (UCMP), Sam McLeod, Vanessa Rhue, Xiaoming Wang, and Jim Dines (LACM), Chris Conroy (UCMVZ), George Corner and Ross Secord (UNSM), Amanda Millhouse (USNM), Darrin Pagnac and Sally Shelton(SDSM), Gregory Wilson Mantilla, Ron Eng, Meredith Rivin, and Chris Sidor (UWBM), Judith Galkin (AMNH), as well as John Wible, Suzanne McLaren, Amy Henrici, and Matt Lamanna (CM). Chris Widga (ETSU) assisted with micro-CT scanning and data processing.

Additional Information and Declarations

Competing Interests

Author Contributions

Data Availability

New Species Registration

The authors declare there are no competing interests.

Joshua X. Samuels conceived and designed the experiments, performed the experiments, analyzed the data, prepared figures and/or tables, authored or reviewed drafts of the article, and approved the final draft.

Jonathan J.-M. Calede conceived and designed the experiments, performed the experiments, analyzed the data, prepared figures and/or tables, authored or reviewed drafts of the article, and approved the final draft.

Robert M. Hunt, Jr. conceived and designed the experiments, performed the experiments, authored or reviewed drafts of the article, collected fossil materials, documented stratigraphic position, and approved the final draft.

The following information was supplied regarding data availability:

Complete cranial, dental, and postcranial measurements of studied specimens are available in Table S3. Nexus file used in phylogenetic analyses, including character states for all studied taxa are available in Data S1 and S2.

The raw data from micro-CT scans including image series and mesh models of the skull and dentary are available at MorphoSource:

- Skull CT Image Series: https://doi.org/10.17602/M2/M460741

- Skull CT mesh: https://doi.org/10.17602/M2/M460708

- Dentary CT Image Series: https://doi.org/10.17602/M2/M468758

- Dentary CT mesh: https://doi.org/10.17602/M2/M468765.

The following information was supplied regarding the registration of a newly described species:

Publication LSID: urn:lsid:zoobank.org:pub:E7B84E29-87D4-4DF5-8CA3-BED1964D3030

Genus name: Aurimys urn:lsid:zoobank.org:act:74FFA744-8369-400C-BF1B-67C3EC0EB710

Species name: Aurimys xeros urn:lsid:zoobank.org:act:EF7FA722-33A2-4FAC-9386-2723BAE5A5BE.

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
