# Peer review of "The earliest dipodomyine heteromyid in North America and the phylogenetic relationships of geomorph rodents"

_PeerJ, doi:10.7717/peerj.14693_

## Round 0.1 · original submission · Major Revisions

I received two reviews of your manuscript, both of which were mostly positive. Reviewer one points out, however, important shortcomings in relation to the graphics used in the ms. You need to improve the graphic support of your work, and this opinion is shared by our reviewer #2. Reviewer #2 also suggested to reformat the Table S1 to allow the readers easily check each taxon’s character states. As you will see, all suggestions made by both reviewers are highly useful and will help you to improve your work.

·

Basic reporting

I would like to express my impossibility to make a complete review of the present manuscript because the inadequacy of graphic support. Authors accompanied a lengthy text (about 40 pages excluding references) with 5 figures but just 1 illustrating anatomical aspects of the fossils studied. The single figure where the new genus and species was illustrated is composed by several views of the cranium, dentary and dentition (occlusal surfaces). The overall quality of this figure is enough to serve as holotype illustration but is poor to follow a detailed anatomical description as the one provided in the main text. More indeed, according to the main text authors conducted a detailed morphology-based phylogenetic analysis including the revision of numerous extant and fossil heteromyids, but there is no a single provided illustration for comparisons. To my best understanding, important paleontological findings must be supported with suitable and abundant graphic material. Otherwise, to the reader (as was a reviewer is) it is impossible (or very hard) to judge authors statements. I really encourage the authors to enrich a preliminary important manuscript with valuable graphic support regarding anatomical issues. Prima facie, the use of widespread of advanced technological facilities is recommended, particularly 3D ct-scan reconstructions, plus drawings where anatomical interpretations are depicted. If ct-scan facilities are not available for the authors, are strongly suggested to photograph the holotype employing not only detailed magnification but also the traditional method of magnesium sublimation. The latter, an easy to obtain “wash,” ensures a uniform response to illumination, a specially recommended technique for rich colored fossils. In addition, I encourage the authors to provide comparable pictures of the other numerous extant and fossil heteromyids studied or mentioned in this study.

Experimental design

nothing to observe in this instance

Validity of the findings

nothing to observe in this instance

Additional comments

nothing to observe in this instance

·

Basic reporting

no comment

Experimental design

no comment

Validity of the findings

no comment

Additional comments

Dear Dr. Abdala,
I really appreciated the opportunity to review Samuels, Calede and Hunt’s manuscript. Of the many manuscripts I have peer-reviewed over the last 34 years, their manuscript is well organized and grammatically one of the best written that I have reviewed.

The authors provide a detailed, but concise, description of a new early dipodomyine and its allocation to Dipodomyinae is well supported by their comprehensive and accurately preformed morphological analyses. Moreover, their analyses provide new, highly significant insights to our knowledge of the phylogenetic relationships of geomorph rodents. Their new proposed phylogenies also point out minor conflicts with the results of certain molecular analyses, highlighting that there are still questions to be answered. Their raw data appears accurate and reliable, and their figures are excellent. Below I made a few comments and suggestions, and noted some lapses between the References section and text/tables. Hopefully, these will help to improve their paper. Considering the overall value of their manuscript to palaeontologists and mammalogists, I strongly recommend publication in PeerJ with minor revision.
Sincerely,
Thomas S. Kelly
Research Associate, Vertebrate Paleontology Department,
Natural History Museum of Los Angeles County

Specific comments regarding corrections needed

Line 158, ..”all but one major families”..... should read .....”all but one major family”....

Line 408, ....”face is...” should read.....”face, is.....

Line 421,.....”a caudal vertebra (Figure 2H).” However, Figure 2H is an occlusal view of the lower dentition. Either add an image of the caudal vertebra to Figure 2 or delete “(Figure 2H)” from sentence.

There are a number of lapses between the citations in the text/tables and the References section that need to be addressed.
1. Citations in text, but not in References section.
Bestland & Retallack, 1994 – line 249
Brusatte et al., 2008 – line 242
Bell & Llyod, 2015 – line 243
Paradis et al., 2004 – line 244
Swofford, 2002 – line 209

2. Citations in References section, but not in the text.
Hay, 1963
Kotler & Brown, 1988 – line 1047
Korth, 1994 – line 1026
Prothero & Rensberger, 1985 – line 1096
Swisher, 1992 – line 1178
Westerhold et al., 2020 – line 1223

2. Citations in Tables that are not included in the References section. I assume that the authors want to include the references cited in tables in the References section because they already did so for some references (i.e., CzaplewskiI, 1990 in Tables 1-3; Rensberger, 1973b in Table 1) and the tables will be part of the online publication (not supplementary files). I tried to add the journals in parentheses that I think the author’s citations refer to in Table 1.
Bertand et al, 2016 – (Proc. Royal Soc. London)
Dowler & Genoways, 1978 - (? Mamm. Sp., ASM)
Galbreath, 1948 - (Publ. Mus. Nat. Hist., Univ. KS)
Galbreath, 1961- (?)
Gazin, 1930 – (Carnegie Inst. Wash. Publ. 404)
Gazin, 1932 - (Contr. Paleo. Carnegie Inst. Wash.)
Kelly & Lugaski, 1999 – (Bull. SCAS)
Korth, 1979 – (Annals Carnegie Mus.)
Korth, 1980 – (JP)
Korth, 1993 – I think this citation is not the same as Korth, 1993 (JM) in References section, which is his paper on Hitonkala. Instead I think this refers to Korth, 2008, his revision of Tenudomys (JVP). If correct, then need to have Korth 1993a and 1993b in the References section.
Korth, 1995 – (JP)
Korth, 2007 – (Paludicola)
Korth, 2008 – reference for Mioperognathus and Eochaetodipus – I think this reference is not the same as Korth, 2008 (Paludicola) listed in the References section. Instead I think this refers to Korth, 2008 (Geodiversitas). If correct, then need to have Korth, 2008a and 2008b in the References section.
Korth, 2013 – (Paludicola)
Korth & Branciforte, 2007 – (Annals Carnegie Mus.)
Matthew, 1910 – (Bull. AMNH)
McDonald, 1970 – (LACM Sci. Ser.)
Reeder, 1960 – (Bull. SCAS)
Shotwell, 1967 - (Bull. 9 Mus. Nat. Hist., Univ. OR)
Souza, 1986 – (?)
Wahlert, 1974 - (Bull. Mus. Comp. Zool.)
Wahlert, 1978 - (Proc. Bio. Soc. Wash.)
Wahlert & Souza, 1988 (AMNH Nov.)
Wilson, 1936 – (Carnegie Inst. Wash. Publ. 473)
Wood, 1932 – (?)

Additional suggestions that could improve the paper, which the authors and editor can determine if they should be implemented.

Text
Line 431, ....”(#9).... I found (#9) a little confusing at first until I realized that it referred to character number 9 in Supplemental Table S4. Perhaps it would be good to clarify at the first reference in the text that #9 refers to the Supplemental Table S4, maybe like (#9, character number in Table S4 here and below). However, there are other options. I understand including large data sets in supplementary files to reduce the size of the published article, but I personally like to have some of that data available in the main paper. Furthermore, many readers do not download those files. In particular, Supplemental_Table_S4, which lists the characters and character states, along with Suppemental_Table_S1, the character state matrix used in the analyses, are very important to other researchers trying to evaluate the character state assignments for each taxon. I would prefer if these were included in the main paper, but with Table S1 formatted so that the reader can easily check each taxon’s character states. In other words, the characters/states columns are aligned. This does not necessarily mean that the Nexus file in Table S4 needs to be eliminated, because in that format, a researcher can copy and paste the file for use in a number of cladistic analysis programs. Having both formats available would be useful and, perhaps, other future researches using the matrix could further refine it by adding new characters and character states when they become available. Whether these proposed changes are needed can be determined by the authors and editor.

line 272, ..”dated to 22.6 ± 0.13 Ma”.... During the 1990’s, the 40Ar/39Ar dates that Carl Swisher ran at the Berkeley Geochronology Center used a date of 27.84 Ma for the Fish Canyon Tuff (FCT) sanidine neutron fluence monitor, which was proposed by Deino and Potts (1990). However, Kuiper et al. (2008) proposed the currently accepted date of 28.201 Ma for the FCT. Unfortunately, the uncorrected dates in Albright et al. (2008) have been repeated in subsequent papers on the John Day Formation (e.g., Fremd, 2010; Korth and Sameuls, 2015). Using the Microsoft Excel application ArArReCalc of McLean (2009), I come up with a corrected date for the “Across the River Tuff” at 22.89 ± 0.13 Ma. Fortunately the corrected date does not affect any of the author’s conclusions, but it would be good if the authors updated any uncorrected 40Ar/39Ar dates cited in their paper. I have solved this problem by just adding a sentence to the Methods and Materials section of my papers that states “older published 40Ar/39Ar dates were recalibrated relative to the new Fish Canyon Tuff sanidine interlaboratory standard of 28.201 Ma.” Perhaps the authors might consider doing the something similar.

Figures
The figures are well prepared, but additional figures could be useful. In particular, it is difficult to see certain cranial characters on Figure 2 that were well described by the authors in the text. Perhaps the authors might consider adding line drawings of the skull pointing out the positions of the foramina, cranial sutures, bones or any other important character states that were used in their cladistic analysis.

Figure 5, “geologic ages of known fossils are represented by thick black lines.” I am a little confused about this phrase. It appears the authors are not referring to the ranges of the fossil species shown in the figure, but the black lines are the geologic range from the divergence times based on the calibration points in the molecular analyses of Hafner et al. (2007). They used a Perognathus specimen (partial skull) from the John Day Formation that James (1963) noted was dentally very similar to P. furlongi, but hesitated assigning it to the species. James (1963:110) also noted “that if more complete skull material of P. furlongi were known it is possible that the two could be separated taxonomically.” To the best of my knowledge, the John Day specimen has never been described, so its specific identity has not been confirmed. Differentiating species of Perognathus based only on dental morphology can be difficult. Hafner et al. (2007) used the John Day specimen, estimated at 22-20 Ma, as their calibration point for Perognathinae, which is acceptable for the subfamily calibration point. Although whether it actually represents P. furlongi is debatable.
The holotype specimen the authors studied, LACM (CIT) 35 is from locality LACM (CIT) 64, which is from the canyon adjacent to Apache Canyon, Caliente Formation, Cuyama Badlands, CA (Clarendonian). To the best of my knowledge, P. furlongi has been otherwise recorded from the early Barstovian (Ba1, Green Hills Fauna and Upper Dome Spring Local Fauna), middle Barstovian (Ba2, Cunningham Hill Fauna), early Clarendonian (Cl1, Matthews Ranch Local Fauna, Iron Canyon Fauna, Doe Spring Local Fauna), late Clarendonian (CL2-3, Ricardo Fauna), and early Hemphillian (Hh1, Dove Spring Fauna) (e.g., James, 1963; Barnosky, 1986; Kelly and Lander, 1988, 1992; Flynn et al., 2008; Prothero et al, 2008; Whistler et al., 2009; Kelly and Whistler, 2014). Thus based on the above records, an estimate for the geochronologic range of P. furlongi appears to be early Barstovian to early Hemphillian (Ba1-Hh1, or about 15.8 – 8.8 Ma). If one accepts referral of the John Day specimen to P. furlongi, it would result in a geochronologic range for the species of ~22-9 Ma. I doubt that a small mammal species with a short generation time ranged through 23 million years unchanged. Because the material referred to the species is often inadequate, as James first noted, it is likely that more complete material from many of the localities would result in the recognition of additional species. This being said, why doesn’t its black line extend up to the early Hemphillian? I know my comments on this are a little too much, but I think it would be good for the authors to just consider further clarifying their phase “geologic ages of known fossils are represented by thick black lines” to the reader.

---

## Round 0.2 · Minor Revisions

I think that we are almost there. A few more suggestions have been sent by our reviewers. Please consider modifying your manuscript accordingly.

·

Basic reporting

Already addressed in the first review.

Experimental design

Already addressed in the first review.

Validity of the findings

Already addressed in the first review.

Additional comments

Dear Dr. Abdala,
The revised version of Samuels, Calede and Hunt’s manuscript is excellent and they addressed all of my prior concerns except one. That is, Galbreath (1948) cited in Table 1 is still missing from the References section. An additional new typo is on line 499 of the revised pdf, where a space is missing between “in” and “Prodipodomys.” Their addition of Figures 3-7 greatly improved the reader’s understanding of the descriptions in the body of the text. With the addition of the missing reference and space, I strongly recommend their paper be accepted for publication in PeerJ with no further changes.
Sincerely,
Thomas S. Kelly

Reviewer 3 ·

Basic reporting

The article is acceptable; the figures are excellent, the discussions, descriptions, and conclusions are very good. I only have a few minor suggested revisions (see below).

Experimental design

no comment

Validity of the findings

Conclusions are sound and well supported.

Additional comments

The paper is well-written and the research and comparative work is excellent and thorough. I only have a few comments regarding terminology and formatting.

1. The authors list the sources of the dental terminology as Wood and Wilson, 1936 and Korth, 1997a. However, in their description of the teeth they use the directional terms "medial and lateral". Neither of these references use these terms for teeth, but rather lingual (=medial) and either labial or buccal (=lateral). These latter terms are standard in the description of rodent teeth, which, in some cases is not standard for other mammals.

2. The authors use the first person (we or our) 53 times throughout the text!! Although grammatically correct it feels like a little bit self-aggrandizement. The authors are clearly listed, and the reader can assume that all conclusions are those of the authors; the reader need not be reminded that the authors made the observations or drew the conclusions.

---

## Round 0.3 · accepted · Accept

Thank you very much for your careful consideration of the reviewer's suggestions. I think that this paper is ready to publish. Nice paper!